# Non-visual photoreceptive brain specification in sea urchin larvae

Junko Yaguchi [1,9], Koki Tsuyuzaki [2,3,4,5,9,10] ✉, Ikutaro Sawada [6], Atsushi Horiuchi[6], Naoaki Sakamoto [7], Takashi Yamamoto [8], Takahiro Yamashita [6] & Shunsuke Yaguchi [1,5,10] ✉

Centralized nervous systems enable animals to detect environmental cues and coordinate behavior, but their evolutionary origins in deuterostomes remain unclear. Among deuterostomes, echinoderms—such as sea urchins—have long been thought to lack brain-like structures, especially in larval stages. Although recent gene expression and neural activity studies suggest brain-like properties in sea urchin larvae, direct links to behavior are still emerging. Here, we identify a light-sensitive cluster of neurons in the posterior neuroectoderm of sea urchin larvae. These neurons express UV-sensitive Opsin5 and regulatory genes such as *rx*, *otx*, *six3*, and *lhx6*, which are conserved in the vertebrate diencephalon. We mapped this domain using single-cell RNA sequencing and in situ hybridization. Knockdown of Opn5L impaired light-dependent swimming, indicating an active role in photoreception. While further work is needed to fully establish circuit-to-behavior relationships, our findings add to growing evidence that sea urchin larvae possess a non-visual photoreceptive neural center with molecular features shared by vertebrate brain regions. This suggests that such domains originated in the deuterostome ancestor and contributed to the early evolution of brain function.

The evolution and acquisition of the brain have undoubtedly played a crucial role in the success and diversification of chordates, including humans. While the origin of the main component of the brain, neurons, remains contentious, evidence from ctenophores—which possess neuron-like cells—suggests that neurons may have originated in the last common ancestor of metazoans. Alternatively, neurons might have independently evolved in the ctenophore lineage after their emergence in the common ancestor of cnidarians and bilaterians[1–3]. Regardless of their exact origin, neurons enable animals to sense internal and external environments, coordinate autonomous body functions, and maintain homeostasis by signaling to other tissues and organs. Among these capabilities, one of the central functions of the brain/central nervous system is to integrate external information and translate it into adaptive motor responses. For animals, the ability to respond to environmental stimuli is fundamental for survival, making the evolutionary origins of the brain a topic of profound biological interest. Debate continues over whether the complex brains of protostomes, such as insects and cephalopods, and vertebrates evolved from a shared ancestral structure or arose independently[4–9].

Within deuterostomes, it is proposed that brain regions derived from the anterior neuroectoderm—specifically the forebrain/pre-oral brain—play a central role in processing environmental stimuli,

[1]Shimoda Marine Research Center, University of Tsukuba, Shizuoka, Japan. [2]Department of Artificial Intelligence Medicine, Graduate School of Medicine, Chiba University, Chiba, Japan. [3]Institute for Advanced Academic Research (IAAR), Chiba University, Chiba, Japan. [4]Laboratory for Bioinformatics Research, RIKEN Center for Biosystems Dynamics Research, Wako, Saitama, Japan. [5]PRESTO, Japan Science and Technology Agency, Tokyo, Japan. [6]Department of Biophysics, Graduate School of Science, Kyoto University, Kitashirakawa-Oiwake, Kyoto, Japan. [7]Graduate School of Integrated Sciences for Life, Hiroshima University, Higashi-Hiroshima, Hiroshima, Japan. [8]Genome Editing Innovation Center, Hiroshima University, Higashi-Hiroshima, Hiroshima, Japan. [9]These authors contributed equally: Junko Yaguchi, Koki Tsuyuzaki. [10]These authors jointly supervised this work: Koki Tsuyuzaki, Shunsuke Yaguchi. ✉e-mail: koki.tsuyuzaki@gmail.com; yag@shimoda.tsukuba.ac.jp

particularly light, as suggested by the chimera hypothesis[10]. These regions exhibit conserved structures across the clade, with visual opsins expressed in superficial eye structures associated with vision, while non-visual opsins are located in deeper brain areas, such as the diencephalon, and later develop into the hypothalamus and the paraventricular organ[11–13]. Recently, a UV-sensitive non-visual opsin, Opsin5, has been identified in vertebrates, showing expression deep within these regions[14–17].

The proboscis of hemichordates, while not a defined "brain" per se, contains photoreceptor-like eye spots and shares gene sets with the vertebrate forebrain, leading to suggestions of a homologous relationship. However, the scattered and non-specific neural patterning in this region does not provide sufficient evidence to confirm an evolutionary link[10,18,19]. Therefore, determining when and how the chordate-type brain emerged and diversified in deuterostome lineage remains difficult, partly due to the lack of a centralized brain structure in

echinoderm larvae, the closest extant relatives to chordates[20] (Fig. 1a). Echinoderm larvae lack a notochord, nerve cord, and any longitudinal axial structures, and exhibit gene expression patterns that do not clearly define an anterior-posterior axis in the ectoderm, complicating efforts to trace the evolutionary origins of the vertebrate brain[20,21]. On the other hand, recent studies have reported not only forebrain-like gene expression profiles in echinoderms[20,22,23], but also the presence of gene regulatory networks that connect these genes[24–26], as well as neural circuits capable of integrating external environmental cues into behavioral outputs[27,28].

Although genomic data from echinoderms have revealed various non-visual opsin–like photoreceptors, functional analyses demonstrating their localized expression in the anterior neuroectoderm during larval stages are still limited[29]. Among non-visual opsins, Opsin3.2 is expressed adjacent to the anterior neuroectoderm in sea urchin larvae[30,31], where it functions in concert with serotonergic

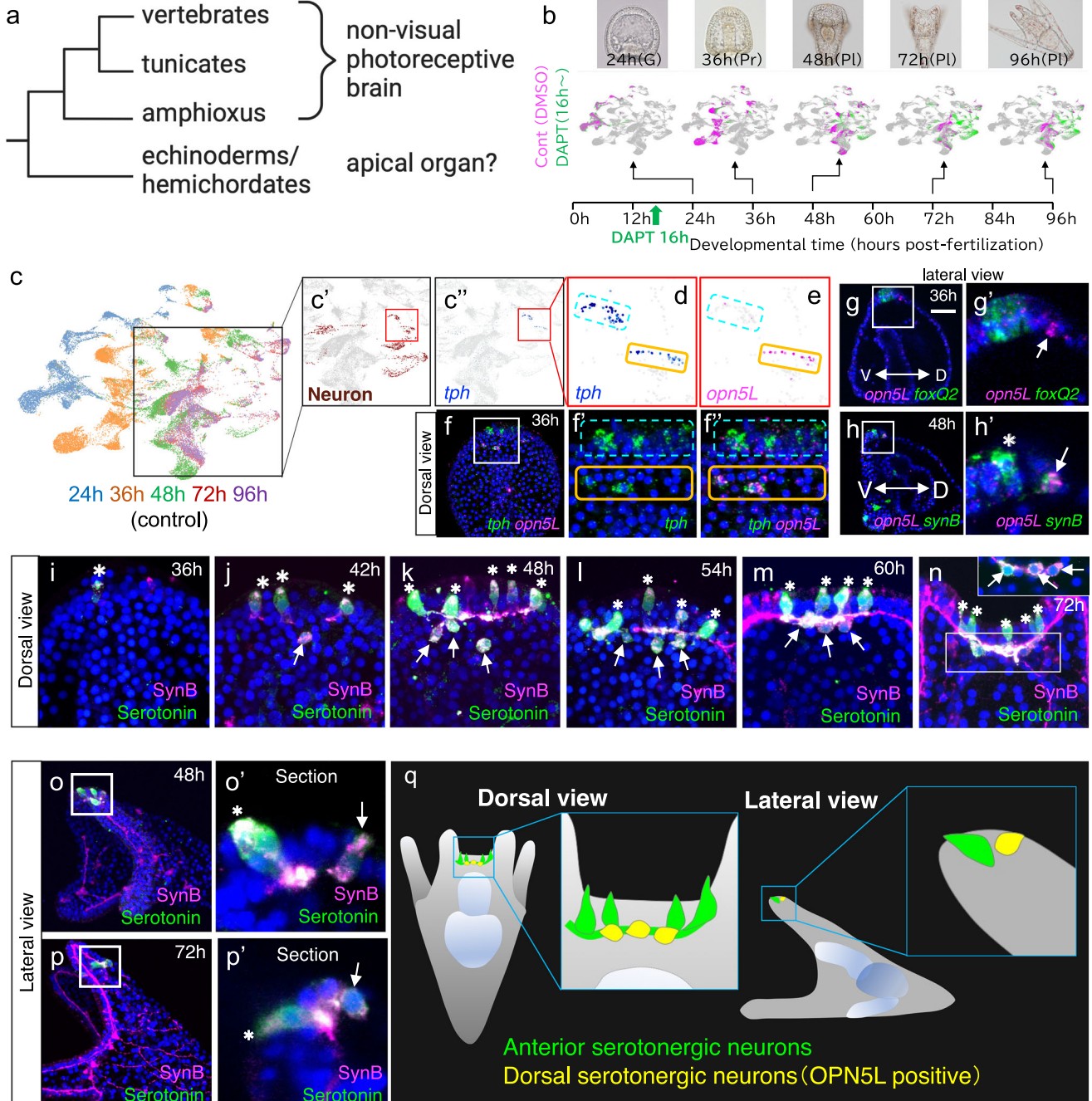

**Fig. 1 | Photoreceptor-positive and -negative serotonergic neurons are present in the brain of sea urchin larvae. a** A simplified phylogenetic tree indicating that non-visual photoreceptive brain structures are present in chordates but remain unclear in echinoderms and hemichordates. **b** Schematic overview of the experimental design for scRNA-seq with DAPT treatment from 24 (gastrula) to 96 hours (pluteus). In the UMAP plot, magenta represents control cells, while green represents DAPT-treated cells. **c–e** A UMAP visualization of scRNA-seq data from control samples at 24 to 96 hours, with different color coding. As explained in Supplementary Fig. 1, all UMAPs presented in this paper represent control-only data derived from the control-DAPT integrated dataset, with DAPT-subtracted. The magnified region within the rectangle in (**c**) is shown in (**c'**), where neurons are highlighted in brown. **c"** Only *tph*-expressing cells are highlighted in blue. *tph*-expressing cells are clustered into two (**d**), while *opn5L* is expressed in only one of them (**e**). Light blue dotted lines, anterior serotonergic neurons; orange lines, dorsal serotonergic neurons. **f.** Expression patterns of *tph* and *opn5L* in 36-hour larvae, viewed dorsally. Cells co-expressing *tph* and *opn5L* are located in the dorsal region (orange-lined rectangle), distinct from the anterior region, where *tph* is expressed alone (light blue dotted-line rectangles). The confocal images **f'** and **f"** correspond to the gene expression regions shown in the UMAP (**d, e**). The box region in **f** is magnified in **f'** and **f"**. **g** Expression patterns of *opn5L* and *foxQ2* mRNA viewed laterally in 36-hour (**g, g'**). Bar = 20 μm. **h** Expression patterns of *opn5L* and *synaptotagminB* (*synB*) viewed laterally in 48-hour larvae (**h, h'**). The magnified views in **g'** and **h'** show that *opn5L*-expressing cells are dorsally positioned, separate from the *foxQ2*-expressing region as well as *synB* cells (asterisk), which represent anterior serotonergic neurons at this stage. Arrows indicate *opn5L*-expressing cells. V, ventral; D, dorsal. Expression patterns of SynaptotagminB (SynB) and serotonin in 36-hour to 72-hour larvae, viewed dorsally (**i–n**) and laterally (**o–p'**). Anterior serotonergic neurons (asterisks) in the larval apical region first appear around 36 hours (**i**). Dorsal serotonergic neurons (arrows) emerge from 42-hour larvae (**j**) and gradually move closer to the anterior serotonergic neurons by 54 hours (**k**). The percentage of dorsal serotonergic neurons detected was 1.5% in 36-hour larvae (*n* = 136), 43.7% in 42-hour larvae (*n* = 126), and 87.3% in 48-hour larvae (*n* = 63). By 72 hours, dorsal serotonergic neurons became difficult to detect without sectioning (**n**). The inset in **n** shows a dorsal section of the central rectangle area in **n**. Lateral views clearly distinguish dorsal serotonergic neurons (arrows) from anterior serotonergic neurons (asterisks) in both 48-hour (**o, o'**) and 72-hour (**p, p'**) larvae. **q** Schematic representation of anterior serotonergic neurons (green) and dorsal serotonergic neurons (yellow), shown in dorsal and lateral views.

neurons to couple light perception with gut activity[27]. Opsin2 is also expressed near the anterior neuroectoderm and regulates light-dependent swimming behavior[32]. In addition, identification of neuropeptides provides evidence that echinoderms may share hypothalamus-like characteristics with vertebrates[31,33,34]. Notably, the neural function appears to be carried out by neurons derived from the anterior neuroectoderm, suggesting that brain-like features may already be present in echinoderm larvae.

Here, we use sea urchin larvae—the most extensively studied echinoderm/ambulacrarian model in neurobiology—to present gene expression and functional data that define a non-visual photoreceptive brain region within echinoderm larvae. Together with recent evidence suggesting that neurons derived from the anterior neuroectoderm are involved in light responses[27,28], our findings add support to the view that some aspects of brain-related organization in chordates may trace back to the common ancestor shared with echinoderms.

## Results
### Single-cell RNA-seq unraveled the presence of non-visual photoreceptive neurons in the anterior neuroectodermal region of sea urchin larvae

The anterior neuroectoderm-derived nervous system in sea urchin larvae primarily consists of serotonergic neurons, positioned along the left-right body axis in early pluteus stages[20,35]. To investigate this region and identify potential unreported photoreceptive systems, we performed single-cell RNA sequencing (scRNA-seq) during neurogenesis in *Hemicentrotus pulcherrimus*. There are certain limitations in using scRNA-seq to capture neuronal properties in relatively late-stage sea urchin larvae. This is because, during the larval stages, when the total number of cells is around 1000 to 1500, specific neurons such as serotonergic neurons are present in extremely low numbers—typically only 4 to 6 cells—raising concerns about whether such rare populations can be adequately analyzed by scRNA-seq. Therefore, based on previous studies, we utilized a strategy involving γ-secretase inhibitor DAPT (N-[N-(3,5-Difluorophenacetyl)-L-alanyl]-(S)-phenylglycine t-butyl ester) treatment, which inhibits Delta-Notch—mediated lateral inhibition. This inhibition disrupts the normal suppression of neuronal differentiation fates and leads to a dramatic increase in the number of neurons[36]. Using the DAPT to enhance neuronal capture, we observed improved neural clustering in the integrated dataset compared to controls, allowing finer resolution of neural cell populations (Fig. 1b, Supplementary Fig. 1). We focused on serotonergic neurons, the principal neurons of the brain-like region that have been the most extensively studied to date. These neurons were identified based on the expression of *tryptophan 5-hydroxylase* (*tph*), the rate-limiting

enzyme in serotonin biosynthesis. Notably, we found that the *tph*-positive cells were clearly separated into two distinct clusters on the UMAP (Fig. 1c, d), which shows only control cells extracted from the integrated dataset of control and DAPT-treated samples, challenging the previous assumption that serotonergic neuron types in the anterior neuroectoderm were poorly defined[37–39]. One cluster uniquely expressed *opn5L* (HPU_23194), a non-visual opsin gene belonging to the neuropsin/Opsin5 group, which we confirmed through in situ hybridization (Fig. 1c–f). In sea urchins, multiple genes have been classified as homologous to Opsin5. However, in this study, Opsin5 specifically refers to the one belonging to the neuropsin group in vertebrates, which is homologous to Opn5L in *Hemicentrotus pulcherrimus* (Supplementary Fig. 2)[29,40]. *opn5L* expression was detected broadly from 12 hr (blastula stage; Bl) to 24 hr (early gastrula stage; EG) and became restricted to several cells by 36 hr (prism stage; Pr) (Supplementary Fig. 3a–h), where it co-localized with neurons slightly dorsal to the *foxQ2* expression domain (Fig. 1g, h). FoxQ2 is an initial specifier of the anterior neuroectoderm. It is broadly expressed throughout the ectoderm at early stages. These findings suggest the presence of a previously unidentified dorsal serotonergic neural population, distinct from anterior serotonergic neurons (Fig. 1q).

Spatial analysis revealed dynamic distribution patterns of serotonergic neurons during development (Fig. 1i–n). By 36 hours post-fertilization, serotonergic neurons were arranged in a single anterior row (Fig. 1i, asterisks). By 42 hours (pluteus stage; Pl), dorsal serotonergic neurons (Fig. 1j, arrows) emerged and seemed to migrate toward anterior neurons, aligning adjacent to them by 54 hours. Anterior serotonergic neurons exhibited an elongated morphology along the anteroposterior axis (Fig. 1i–n, asterisks), while dorsal neurons maintained a spherical shape (Fig. 1j–n, arrows). These distinct morphologies persisted until 72 hours, when dorsal neurons became obscured by overlapping axons but remained identifiable in transverse sections (Fig. 1m–p). Together, these findings reveal two serotonergic neuron populations in pluteus larvae: elongated anterior neurons and dorsal neurons expressing *opn5L* (Fig. 1q). The discovery of *opn5L*-expressing neurons provides novel insights into the diversification of serotonergic neurons and highlights the sea urchin as a model for understanding the function of non-visual opsin expressed in the anterior neuroectodermal region and neural evolution.

### Opn5L cells express genes associated with development of neuroectoderm and brain in other organisms

Based on the scRNA-seq data, we compared the gene expression profiles of all cells within the two *tph*-expressing clusters: one corresponding to dorsal serotonergic neurons (orange-lined rectangle in

Fig. 1d, e) and the other to anterior serotonergic neurons (light blue-dot-lined rectangle in Fig. 1d, e). Among the genes enriched in the dorsal *opn5L*-positive cluster, we excluded hypothetical proteins, general cytoskeletal components, and those with unknown function, and prioritized candidates with a high likelihood of involvement in neurogenesis or neuronal function. These included transcription factors, signaling molecules, and membrane proteins, many of which have been previously implicated in neural development, e.g., refs. 6,8,41,42, (highlighted in grey in Table 1) and validated their expression patterns through in situ hybridization.

Among all the candidate genes analyzed, all except one—which could not be cloned—were detected in the dorsal region where *opn5L* is expressed, at least in DAPT-treated embryos. Since DAPT treatment increases the number of neurons, it enhances the visibility of genes that are otherwise rare in control embryos. Notably, the expression of *z167*, previously reported in *Strongylocentrotus purpuratus* to be involved in serotonergic neuron differentiation[43], ranked highly in the scRNA-seq analysis and was clearly co-localized to *opn5L*-positive dorsal serotonergic neurons through in situ hybridization (Fig. 2a). Although *z167* has been previously reported to be expressed in serotonergic neuronal precursors in *S. purpuratus*[43], based on scRNA-seq data and the spatial analysis of its expression using double in situ hybridization with *foxQ2*[44], our findings suggest that in *H. pulcherrimus*, *z167* is not expressed in anterior serotonergic neurons but rather exclusively in dorsal serotonergic neurons (Fig. 2b, c, g). In contrast, *ebf3*, a multiply functioning transcription factor in vertebrates[45], was found to be exclusively expressed in anterior serotonergic neurons (Fig. 2d, g, Supplementary Fig. 4a). The anterior neuroectoderm specifier FoxQ2 is localized to the most anterior region by 36 hours, and it is known that serotonergic neurons differentiate at its dorsal boundary[38]. However, *z167*-expressing cells at 36 hours are not in close proximately to the *foxQ2* domain; instead, they differentiate in a distinct region located more dorsally than the *foxQ2* area. In contrast, cells expressing *ebf3* are positioned much closer to the dorsal boundary of the *foxQ2* domain. Comparative expression analysis between *ebf3* and *z167* revealed a clear spatial separation along the anterior-posterior axis (Fig. 2e). Double in situ hybridization demonstrated that while the two genes are expressed in close proximity, they are strictly confined to distinct cell populations, especially in DAPT-treated embryos (Fig. 2f). Similarly, in the scRNA-seq UMAP representation of the *tph*-expressing domain, *ebf3* was confined to one cluster (light blue-dot-lined), while *z167* was restricted to another (orange-solid-lined), confirming their mutually exclusive expression patterns (Fig. 2g). Notably, *z167* appears to be a highly specific and persistent marker for dorsal serotonergic neurons, suggesting its potential as a definitive marker for this cell population (Fig. 2h-j, arrows). The UMAP in Fig. 2g represents the same region as shown in Fig. 1d and e. Both the light-blue dashed and yellow solid regions express tph, but they are distinct in gene expression: the light-blue dashed region is *ebf3*⁺/*opn5L*⁻/*z167*⁻, while the yellow solid region is *ebf3*⁻/*opn5L*⁺/*z167*⁺. This mutually exclusive gene expression pattern is supported by the in situ hybridization results in Fig. 2. Although previous studies in *S. purpuratus* reported that *z167* knockdown led to near-complete loss of serotonergic neurons[43], this interpretation conflicts with the observed expression patterns in this study. To investigate this further, we generated Z167 morphants in *H. pulcherrimus* and analyzed the serotonergic neuron formation. In Z167 morphants, serotonin production was abolished exclusively in dorsal serotonergic neurons, while anterior serotonergic neurons remained largely unaffected (Fig. 2k, l). This was further confirmed by experiments using a second, non-overlapping morpholino (Supplementary Fig. 4b). On the other hand, neural differentiation in the dorsal domain appeared intact, as indicated by the clear presence of the pan-neuronal marker SynB (Fig. 2l, arrows, Supplementary Fig. 4c, 4d). These findings indicate that while neuronal differentiation occurs, Z167 is crucial

**Table 1 | List of genes highly expressed in the dorsal serotonergic neuron cluster**

| Order of highly expressed gene | Gene name | Gene ID (HPU) |
|---|---|---|
| **1** | **Hp-Pura-like** | **HPU_23258** |
| 2 | Hp-Hypp-0675 | |
| 3 | Hp-Hypp-4026 | |
| 4 | Hp-Hypp-2319 | |
| **5** | **Hp-Z167** | **HPU_10494** |
| **6** | **Hp-Hmx** | **HPU_02863** |
| 7 | Hp-Hypp-0957 | |
| 8 | Hp-Abtb2-like | |
| 9 | Hp-Qsk | |
| 10 | Hp-Pdzrn4 | |
| 11 | Hp-Salmfap | |
| 12 | Hp-Fut8-1-like | |
| **13** | **Hp-Pde9a-like** | **HPU_24271** |
| **14** | **Hp-Cnpy4** | **HPU_05237** |
| 15 | Hp-Npr1-6 | |
| **16** | **Hp-Ets1/2** | **HPU_15019** |
| 17 | Hp-Gucy2g | |
| 18 | Hp-Prxdxn-like | |
| 19 | Hp-Ncor2 | |
| 20 | Hp-Hypp-0800 | |
| 21 | Hp-Slc34a2-2 | |
| 22 | Hp-Radil-like | |
| **23** | **Hp-Six3** | **HPU_00364** |
| 24 | Hp-Delta | |
| 25 | Hp-Txndc16 | |
| 26 | Hp-Fchsd2 | |
| **27** | **Hp-Sna** | **HPU_14572** |
| 28 | Hp-Ube2q1L | |
| 29 | Hp-Gi | |
| 30 | Hp-Phldb1L | |
| 31 | Hp-Iqce-like | |
| **32** | **Hp-Otx** | **HPU_04624** |
| 33 | Hp-Pgrp2-like | |
| 34 | Hp-Lrr/Igr-12 | |
| 35 | Hp-Hypp-0172 | |
| 36 | Hp-P53L | |
| 37 | Hp-Pde9a-1 | |
| 38 | Hp-Rergl2 | |
| 39 | Hp-Btk-like | |
| 40 | Hp-Hypp-7235 | |
| **41** | **Hp-Opn5L** | **HPU_23194** |
| 42 | Hp-Oprm1L-like | |
| 43 | Hp-Galr2L-23 | |
| 44 | Hp-Pp4r1 | |
| **45** | **Hp-Awh** | **HPU_17956** |
| 46 | Hp-Ksr2 | |
| **47** | **Hp-Pax1-9** | **HPU_13853** |
| **48** | **Hp-Rx** | **HPU_04689** |
| 49 | Hp-Bcdo2-like | |
| 50 | Hp-Acbd3L | |
| | ⸱ | |
| **191** | **Hp-Cry1** | **HPU_09510** |
| **241** | **Hp-Nkx2.1** | **HPU_00866** |

Genes highlighted in bold were validated by in situ hybridization or immunohistochemistry.

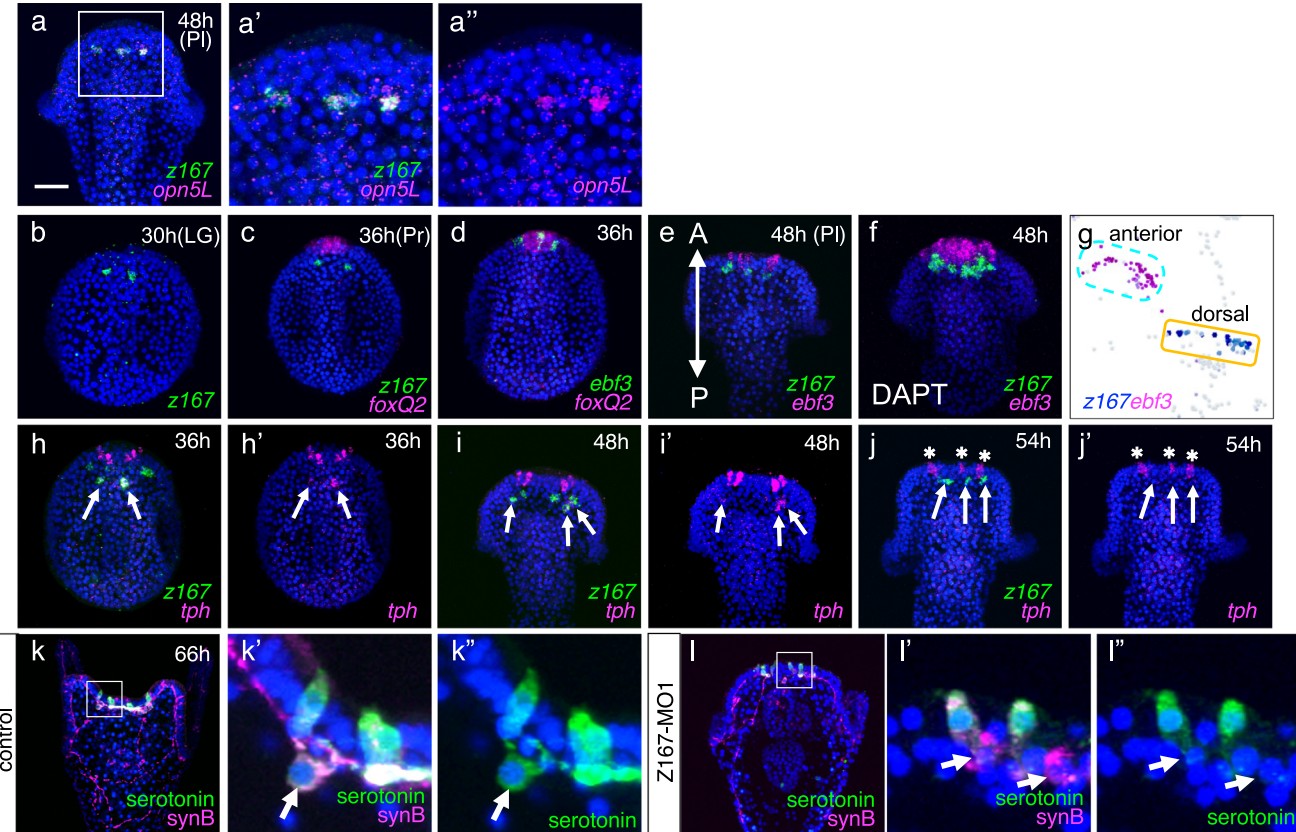

**Fig. 2 | Gene expression patterns and function of Z167 in dorsal serotonergic neurons, in which *opn5L* is expressed. a** Co-expression of *opn5L* and *z167* at 48-hour pluteus larva. Bar = 20 μm. **a', a''** Magnified image of a square in (**a**). **b** Expression pattern of *z167* in 30-hour late gastrula (LG). Detection of *z167* began around 30 hours, with 43.3% of larvae (n = 203) showing *z167* signals. **c** Co-expression pattern of *z167* and *foxQ2* in 36-hour prism larvae. **d** Co-expression pattern of *z167* and *ebf3* in 36-hour larvae. Co-expression pattern of *z167* and *ebf3* in 48-hour pluteus larvae under control conditions (**e**) and with DAPT treatment (**f**). A, anterior; P, posterior. **g** UMAP representation of *z167*- and *ebf3*-expressing cells. **h–j**. Co-expression pattern of *z167* and *tph* in 36- to 54-hour larvae. Expression of *tph* in dorsal serotonergic neurons was transient, and the *tph* signal was nearly undetectable in 54-hour larvae (**j, j'**, arrows). The number of *tph*-positive cells in the

dorsal region was 1.53 ± 0.22 (N = 3, n = 53, 27, 7) in 48-hour larvae and 0.58 ± 0.24 (N = 3, n = 10, 14, 15) in 54-hour larvae. In contrast, *z167* expression persisted beyond 54 hours, suggesting that the *z167* probe serves as a reliable marker for dorsal serotonergic neurons. The number of *z167*-positive cells was 3.1 (n = 39), with 100% detection in 54-hour larvae. Arrows indicate dorsal serotonergic neurons, and asterisks indicate anterior serotonergic neurons. **k, l** Expression patterns of serotonin and Synaptotagmin B (SynB) in control larvae (**o**) and z167 morphants (**l**). In z167 morphants, SynB (neuronal marker) was detected in the region where dorsal serotonergic neurons are typically found, whereas serotonin signals were absent (**l–l''**). The boxed regions in **k** and **l** are magnified in **k', k''** and **l', l''**, respectively. Arrows indicate dorsal serotonergic neurons.

for serotonin synthesis and possibly maintenance only in dorsal serotonergic neurons.

Additionally, *snail* (*sna*), known to be expressed and function in migratory cells during neurogenesis in other systems[46,47], was also prominently detected in dorsal serotonergic neurons (Fig. 3a). Our experiments revealed that *hmx*, a gene known to be expressed in cranial ganglia in vertebrates, is expressed near the anterior region in sea urchins for the first time[48] (Fig. 3b). Similarly, *six3*, previously reported to exhibit widespread expression in the oral region after 24 hr (Bl) of sea urchin development[22], was newly identified to be expressed in dorsal serotonergic neurons (Fig. 3c). *limc1* (*lhx2/9*) and *awh* (*lhx6*) were also expressed in regions corresponding to dorsal serotonergic neurons (Fig. 3d, e). *otx*, which is known to be expressed in the diencephalon in vertebrates, was broadly detected in the dorsal region, including the dorsal serotonergic neurons (Fig. 3f). The expression of *cryptochrome-1* (*cry1*), implicated in circadian rhythms in sea urchins[49] and in the relationship with Opsin5 in frog[50], was also confirmed (Fig. 3g). The transcription factor *retinal homeobox* (*rx*), typically expressed in various neural progenitors in vertebrates, was also detected in *opn5L/z167*-expressing cells at this developmental stage (Fig. 3h). In early pluteus larvae, *rx* has been reported to localize to *opsin3.2*-expressing cells bilaterally in the anterior region[30,51,52], and

that is confirmed in *H. pulcherrimus* as well (Supplementary Fig. 5a–c). in situ hybridization pattern of other genes highlighted in Table 1 are shown in Supplementary Fig. 5d–h, confirming that all genes are expressed around the dorsal serotonergic neural region. Some of these genes were undetectable in normal larvae due to their low expression levels, but became clearly detectable in the dorsal region of the neuroectoderm in DAPT-treated embryos (Supplementary Fig. 5).

## Dorsal photoreceptor/serotonergic neurons migrate to form the complete nervous system

The dorsal serotonergic neurons express the transcriptional repressor *sna*, a critical factor for migratory neural crest cells in vertebrates[53–56] and migratory mesodermal cells in sea urchins[46,57] (Figs. 3a, 4a–d, Supplementary Fig. 6a–c). To investigate their migratory behavior, we microinjected a DNA construct encoding the fluorescent protein Venus under the control of the z167 cis-regulatory element into sea urchin eggs since *z167* and *sna* are exclusively co-expressed in the dorsal serotonergic neurons. Live imaging at the prism stage revealed that dorsal serotonergic neurons initially located in the dorsal ectoderm migrate over a few hours to the region containing anterior serotonergic neurons, where cell density is notably high (Supplementary Movie 1, Supplementary Fig. 6d). To quantify this migration, we

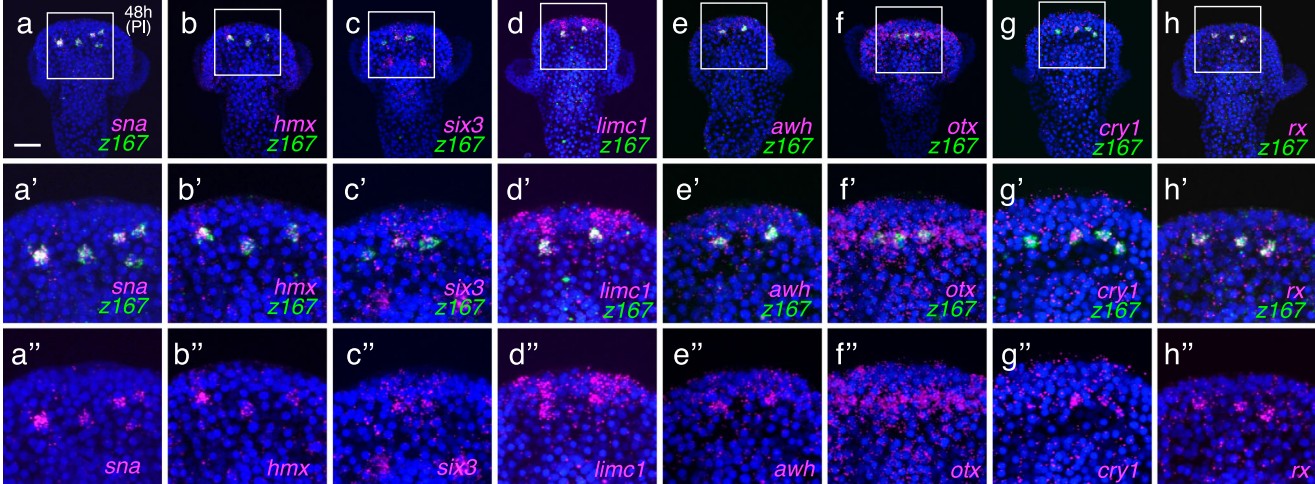

**Fig. 3 | Co-expression genes in dorsal serotonergic neurons with *z167/opn5L*.** **a–h** Co-expression patterns of *sna*, *hmx*, *six3*, *awh* (*lhx6*), *otx*, *cry1*, and *rx* with *z167*, which exhibited relatively high expression levels in dorsal serotonergic neurons based on single-cell analysis in 48-hour pluteus larvae treated with DMSO (control). All factors were detected in the *z167*, i.e., *opn5L*-expressing cells. **a-a"** Co-expression pattern of *sna* with *z167* in 48-hour larvae. Bar = 20 µm. **b-b"** Co-expression pattern of *hmx* with *z167* in 48-hour larvae. **c-c"**. Co-expression pattern of *six3* with *z167* in 48-hour larvae. **d-d"** Co-expression pattern of *limc1* with *z167* in 48-hour larvae. **e-e"** Co-expression pattern of *awh* with *z167* in 48-hour larvae. **f-f"**. Co-expression pattern of *otx* with *z167* in 48-hour larvae. **g-g"** Co-expression pattern of *cry1* with *z167* in 48-hour larvae. **h-h"** Co-expression pattern of *rx* and *z167* in 48-hour larvae. **a'-h"** Magnified image of boxed area in (**a–h**), respectively. All experiments were independently confirmed in more than 3 batches.

measured the distance, i.e. counted the number of cells between *tph*-positive/*z167*-negative anterior serotonergic neurons and *z167*-positive dorsal serotonergic neurons from the samples we showed in Fig. 2j–n and confirmed that the distance was reduced (Fig. 4e, f). Detailed optical sectioning showed that the migrating dorsal serotonergic neurons delaminate into the blastocoel, as evidenced by their absence from the epithelial layer and their penetration through the basal lamina (Fig. 4g). This was further supported by double staining for serotonin and Epith-2, a lateral cell membrane marker[58], which confirmed that these neurons had exited the epithelial layer entirely (Fig. 4h).

As previously reported, migration of primary mesenchyme cells was delayed in Sna-MO1 and MO2 morphants, and some larvae developed into pluteus-stage larvae despite the delay[46] (Supplementary Fig. 6e–m). Based on the presence of a tripartite gut, an open mouth, and elongation of skeletal rods in the pluteus configuration, we judged that the morphants had eventually reached the pluteus larval stage, albeit with some developmental delay, and proceeded with the following observations. In Sna-MO1 morphants, the expression pattern of *ebf3* remained largely unchanged compared to the control, whereas the number of *z167*-expressing cells was reduced. This is supported by the results using non-overlapping morpholino Sna-MO2 (Supplementary Fig. 6n, q, t). Furthermore, at 72 hours, many pluteus-stage larvae lacked detectable dorsal serotonergic neurons (Supplementary Fig. 6n–v). Although we could not directly assess the role of Sna in neuronal migration due to developmental perturbations at early stages in knockdown experiments (Supplementary Fig. 6), the expression of *sna* in migratory neuroectodermal cells of vertebrates and migratory mesodermal cells of sea urchins strongly suggests its involvement in this process. Our data provide the possibility that sea urchin neurons exhibit migratory behavior—an essential feature of nervous system development previously thought to be exclusive to chordates among deuterostomes[56,59,60]. The recognition of migratory neurons as a shared deuterostome trait suggests that this feature was already present in their common ancestor. Intriguingly, the dorsal photoreceptor/serotonergic neurons studied here share gene expression profiles with the vertebrate diencephalon, which migrates during development to form the hypothalamus, pituitary gland, and their related organs[61–63]—key components of the non-visual photoreceptor-associated brain,

e.g., refs. 15,50. These findings suggest a deep evolutionary link between the migratory mechanisms of echinoderm and vertebrate neurons, shedding light on the origins of central nervous system development across deuterostomes.

## Opn5L is required for swimming behavior in sea urchin larvae

The dorsal photoreceptor neurons in sea urchin larvae share similarities with the vertebrate diencephalon, particularly in their expression of Opsin5. While Opsin5 in birds is linked to reproduction[17], its function in sea urchin larvae also appears distinct, as these larvae do not produce eggs or sperm at this stage. To investigate its role, we knocked down Opn5L using morpholino antisense oligonucleotides (MO), which did not affect the growth and the morphology of pluteus larvae and the developmental pattern of the anterior and dorsal serotonergic neurons (Fig. 5a, b), and examined larval behavior under light exposure. Continuous photoirradiation was observed to block floating/swimming behavior in control larvae, leading us to hypothesize that Opn5L mediates this response. This is because other opsins expressed during larval stages have been studied, and none are linked to such behaviors[27,28,32]. In control larvae, approximately 70% sank to the bottom of the culture dish after three days of continuous light exposure, whereas Opn5L morphants maintained their floating behavior (Fig. 5c). This was further confirmed by experiments using a second, non-overlapping morpholino (Supplementary Fig. 7a, b). Although Opn5L might be most effective under UV light, we used white light to avoid UV toxicity. Given that Opsin5 in other organisms responds to violet light[14], our findings strongly suggest that Opn5L in sea urchin larvae detects violet and/or blue light in white light, influencing swimming behavior. This is supported by in vitro photoreactive experiments of sea urchin Opn5L (Fig. 5d, Supplementary Fig. 7c). Broad absorption spectra of sea urchin Opn5L proteins allow them to detect violet and blue light in addition to their maximal sensitivity to UV light like vertebrate Opsin5 proteins20,21.

Previous studies have shown that serotonergic neurons regulate larval swimming behaviors, including ciliary beating, and that serotonin is essential for anti-gravity swimming or floating[37]. In our experiments, control larvae exposed to continuous light exhibited sinking behavior, whereas Opn5L morphants maintained floating. This suggests that light activates Opn5L, enabling larvae to avoid prolonged

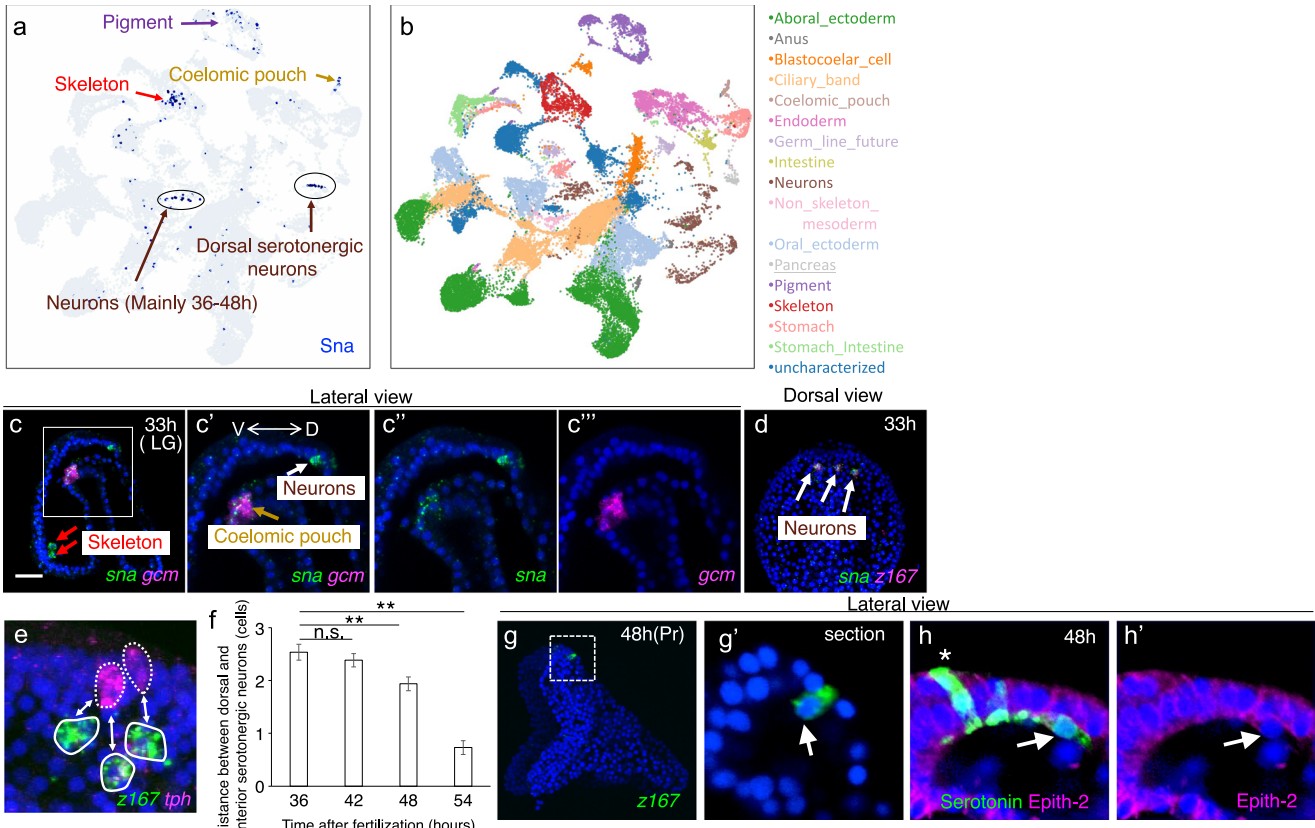

**Fig. 4 | Potential Snail-mediated migration of dorsal serotonergic neurons toward the anterior region. a, b** UMAP representation of *snail* (*sna*) expression. *sna* was expressed in the skeleton, pigment cells, coelomic pouch, and the dorsal serotonergic neurons. **c** Co-expression pattern of *sna* and *gcm* (a marker for secondary mesenchyme cells) in 33-hour larvae. These images clearly demonstrate that *sna* is expressed in the skeleton (red arrows), coelomic pouch (ocher arrow), and neurons (white arrow). V, ventral; D, dorsal. Bar = 20 μm. The boxed area in (**c**) is magnified in **c'- c'''**. **d** Co-expression pattern of *sna* and *z167* in 33-hour larvae, viewed dorsally. *sna* expression in the dorsal ectoderm completely overlapped with the *z167* signal (white arrow). **e, f** Quantification of the spatial relationship between dorsal and anterior serotonergic neurons. In the co-expression pattern of *z167* and *tph*, *z167*-positive cells were identified as dorsal serotonergic neurons, whereas *tph*-only cells were classified as anterior serotonergic neurons (**e**). The number of cells between these two neuronal populations was manually counted in confocal laser microscopy sections. The graph (**f**) shows that from 48-hour larvae, dorsal and anterior serotonergic neurons began to move closer together. The proportion of

dorsal serotonergic neurons in direct contact with anterior serotonergic neurons was 0% at 36 hours ($n = 56$), 1.5% at 42 hours ($n = 65$), 11.1% at 48 hours ($n = 81$), and 51.9% at 54 hours ($n = 52$). The mean number of cells located between dorsal and anterior serotonergic neurons was: 36-hour larvae: 2.54 ± 0.15 SEM, 42-hour larvae: 2.38 ± 0.13 SEM, 48-hour larvae: 1.94 ± 0.13 SEM, and 54-hour larvae: 0.73 ± 0.13 SEM. A one-way ANOVA followed by Tukey's post hoc test was used for statistical analysis. Error bars indicate S.E. ** [36 h vs 48 h], $p = 0.009274$, [36 h vs 54 h], $p = 0.0010053$; n.s. = not significant [36 h vs 42 h], $p = 0.8565991$. **g, h** Dorsal serotonergic neurons were detected on the inner surface of the ectodermal layer. In lateral views, *z167*-expressing cells were occasionally observed on the inner side of the ectodermal layer, as indicated by the arrow in (**g'**). The boxed region in **g** is magnified in **g'**. Similarly, serotonin-expressing cells were also found attached to the inner surface of the ectodermal layer (arrows in **h, h'**). These cells lacked Epith2 signal, an epithelial cell surface molecular marker. Asterisks indicate anterior serotonergic neurons.

exposure, particularly to UV light. In vertebrates, although UV-sensing Opsin5 is expressed in the eyes and can contribute to UV avoidance, an ancestral feature that likely predates modern eyes. Supporting this hypothesis, in Medaka, UV-sensing Opsin5 expressed in the pituitary regulates pigmentation[64]. These findings imply that UV avoidance mechanisms required integration of sensory input into complex behaviors, which may have influenced the evolution of nervous systems. Interestingly, this light-avoidance behavior was observed even under weak light conditions. Under stronger light, previously reported Opsin2-mediated regulation of ciliary beating may dominate, albeit for limited durations[32]. Future studies that integrate these findings will be pivotal for understanding light-driven swimming behaviors, such as diel vertical migration (DVM), in echinoderms and other marine organisms like zooplankton[65–67].

## Discussion

While echinoderms may not exhibit centralized neural structures equivalent to those of chordates, our findings, together with previous

data[27,28,31,33,34], suggest the presence of a coordinated neural domain that integrates environmental light cues and modulates larval behavior. This may reflect an evolutionarily derived form of neural organization that serves brain-like functions in the context of deuterostome diversity[7,10,68]. By identifying a non-visual photoreceptive brain-like region in sea urchin larvae, we reveal that the gene expression profiles in this region share remarkable similarities with those of the vertebrate forebrain. Notably, region-specific transcription factors such as *rx*, *otx*, *nkx2.1*, and *awh* (*lhx6*) are conserved in photoreceptor-, i.e., Opsin5-associate brain area[10] (Fig. 5e, Supplementary Fig. 8). Furthermore, *fez*, a key marker of anterior brain development in vertebrates, is expressed in a comparable anterior domain in sea urchins[69], suggesting that the ancestral deuterostome likely possessed a rudimentary brain-like region capable of integrating environmental stimuli, such as light, into behavioral outputs[27,28]. Unlike the internalized brain of chordates, echinoderm larvae integrate external sensory information within a single ectodermal layer, where neural and epithelial cells have to coexist in a structurally coordinated

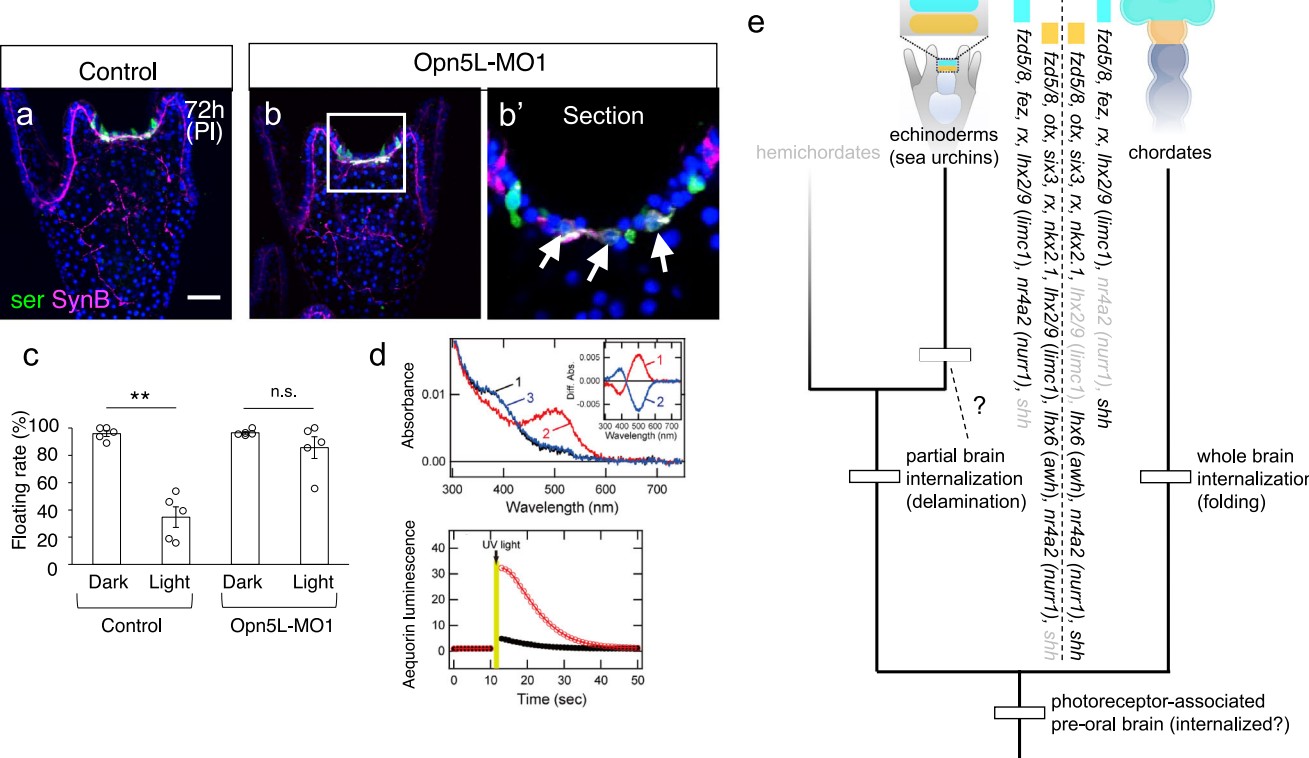

**Fig. 5 | Opn5L function and the potential evolutionary scenario for non-visual photoreceptive brain regional specification. a, b** Expression patterns of serotonin and Synaptotagmin B (Syn B) in control and Opn5L-MO1-injected larvae. Bar = 20 μm. Opn5L-MO1-injected larvae developed into pluteus larvae similar to controls, with both anterior and dorsal serotonergic neurons observed normally (**b, b'**). The boxed region in (**b**) is magnified in (**b'**). Arrows indicate dorsal serotonergic neurons. **c** Differences in floating rates between control and Opn5L-MO1-injected larvae under dark and light conditions. Control and Opn5L morphants were cultured in light or dark conditions from 24 hours post-fertilization, and the number of actively swimming larvae (without sinking) was counted at the four-day larval stage. Control larvae continued to swim in the dark, but their swimming rates decreased significantly under light conditions. In contrast, Opn5L morphants maintained high swimming rates under both dark and light conditions. Photon flux density was set to 150–200 μmol m$^{-2}$ s$^{-1}$. The mean swimming rate in control larvae was 95.9% ± 2.1% SEM ($n = 56, 33, 27, 49, 58$) in the dark and 34.7% ± 7.5% SEM ($n = 38, 41, 51, 32, 59$) in the light. Opn5L-MO1-injected larvae showed a mean swimming rate of 96.5% ± 0.9% SEM ($n = 22, 19, 22, 24, 60$) in the dark and 85.7% ± 8.0% SEM ($n = 48, 58, 38, 18, 61$) in the light. **, $p = 0.0007504$; n.s., not significant, $p = 0.2487$. **d** Molecular property of *H. pulcherrimus* Opn5L protein. (upper panel) Spectral property of *H. pulcherrimus* Opn5L protein. Absorption spectra of Opn5L protein purified after the addition of 11-*cis* retinal were recorded in the dark (curve 1), after UV light (360 nm)

irradiation (curve 2) and after subsequent yellow light (>500 nm) irradiation (curve 3). UV light irradiation shifted the spectrum from the UV region (~390 nm) to the visible region (~500 nm), and subsequent yellow light irradiation recovered the absorbance in the UV region. Combined with the results for *S. purpuratus* Opn5L protein (see Supplementary Fig. 7), sea urchin Opn5L is a UV-sensitive bistable opsin like vertebrate Opsin5. (inset) Spectral changes of *H. pulcherrimus* Opn5L protein caused by UV light irradiation (curve 1) and subsequent yellow light irradiation (curve 2). (lower panel) Opn5L-evoked elevation of intracellular Ca$^{2+}$ level. The Ca$^{2+}$ level change in *H. pulcherrimus* Opn5L-transfected (red curve) or mock-transfected (black curve) HEK293 cells was measured using an aequorin-based luminescent assay. Data are presented as the means of two independent experiments. A transient elevation of the luminescence by light irradiation from UV LED light (365 nm) was observed in Opn5L-expressing cells. This means that *H. pulcherrimus* Opn5L triggers the intracellular Ca$^{2+}$ signaling in a light-dependent manner like vertebrate Opsin5. **e** Conserved gene expression patterns (see Supplementary Fig. 8) in the non-visual photoreceptive brain region between echinoderms and chordates. Gray text indicates genes that are not expressed. While the process and/or pattern of neuronal internalization differs between the two groups, this variation may be linked to differences in the complexity of brain development and evolution. Light blue, future telencephalon (-like); orange, future diencephalon (-like).

---

manner. This organization is precisely governed by Delta-Notch lateral inhibition[70], which ensures the balanced allocation of cells, with some differentiating into neurons while others remain as epithelial or stem-like progenitors. On the other hand, the expression of the vertebrate telencephalon-specific *lhx2/9* (*limc1*)[71] in the dorsal serotonergic neurons of sea urchin larvae suggests that the brain patterning in chordates/vertebrates and echinoderms is not entirely conserved. Moreover, the absence of *shh*, a key regulator of brain development in chordates, may have contributed to the simpler structural organization in echinoderms[42]. However, the presence of conserved transcriptional programs suggests a shared ancestral basis for neural patterning across deuterostomes (Fig. 5e).

The differences between echinoderms and vertebrates can be understood in the context of evolutionary divergence rather than complete novelty. While vertebrates developed internalized, modular

brain structures, such as the hypothalamus, pituitary gland, and telencephalon, echinoderms retained external photoreceptor neurons to meet the ecological demands of their planktonic environment. This divergence reflects an adaptive balance between shared developmental potential and lineage-specific constraints. Interestingly, sea urchin larvae show limited neural internalization through delamination (Fig. 5e), and the nervous systems of adult echinoderms are internalized within their endoskeleton[21]. These observations suggest that mechanisms for neural internalization were likely present in the ancestral deuterostome and were subsequently adapted to meet the needs of different lineages[9]. Rather than representing a unique departure, the neural architecture of echinoderms demonstrates how a shared developmental toolkit can give rise to diverse evolutionary outcomes.

Both the adult and larval nervous systems of echinoderms exhibit features that align with deuterostome neural organization, suggesting

the retention of ancestral patterning despite extensive morphological divergence. The adult echinoderm nervous system, while pentaradial, retains molecular features consistent with anteroposterior axis patterning seen in bilaterians, with conserved anterior and posterior gene expression domains distributed along the ambulacral midline and lateral regions, respectively[21]. At the same time, the larval nervous system maintains a bilaterian-like organization, particularly in the anterior neuroectoderm, where photoreceptor-associated neurons and key transcription specifiers are expressed. This suggests that, at least in the non-visual photoreceptive brain region, sea urchin larvae share organizational similarities with the ancestral deuterostome nervous system. Determining which stage−larval or adult−better represents the ancestral deuterostome neural architecture remains challenging. However, given that both stages exhibit conserved features, echinoderms likely retain fundamental aspects of the deuterostome nervous system, including a brain region homologous to those found in chordates. Importantly, when considering body axis formation, the anterior neuroectoderm-derived brain pattern in sea urchin larvae strongly supports the presence of a non-visual photoreceptive brain structure in the common ancestor of Ambulacraria and chordates. This alignment with body axis patterning further reinforces the idea that the larval nervous system provides critical insights into the ancestral organization of deuterostome neural structures, particularly in defining the spatial identity of the non-visual photoreceptive brain[10].

Finally, the anterior neuroectoderm of sea urchins, where photoreceptor neurons are maintained, highlights their conserved capacity for environmental integration. The discovery of migratory neurons expressing Opsin5 and their role in light-sensitive behaviors underscores the common genetic and developmental frameworks underlying neural systems in deuterostomes. Notably, recent studies have begun to uncover the functions of non-visual Opsin5 in vertebrates[14,15,17,50,64], further emphasizing its conserved and ancestral role in mediating light-responsive behaviors. As sea urchins lack a visual system like eyes, they provide a unique model for investigating the origins and functions of non-visual opsins, offering insights into how these molecules contribute to light sensing and environmental adaptation. While the ancestral deuterostome likely had the genetic potential for complex brain structures, different lineages have elaborated or constrained these features according to ecological pressures and developmental constraints. These findings position sea urchins as an important model for understanding both the shared and lineage-specific aspects of neural and brain evolution, as well as the evolution and functional diversification of non-visual opsins in deuterostomes[11,27,28,32,72].

## Methods

### Animal collection and embryo/larva culture
Adults of *Hemicentrotus pulcherrimus* were collected around Shimoda Marine Research Center, University of Tsukuba, around the Marine and Coastal Research Center, Ochanomizu University, and the Research Center for Marine Biology, Tohoku University. Adult sea urchins were collected under the special harvest permission of prefectures and Japan Fishery cooperatives. Gametes were collected by the *intrablastocoelic* injection of 0.5 M KCl, and the embryos/larvae of *H. pulcherrimus* were cultured at 15 °C in glass beakers or plastic dishes that contained filtered natural seawater (FSW) with 50 µg/ml of kanamycin. 3,5-Difluorophenylacetyl-L-alanyl-L-S-phenylglycine T-butyl ester (DAPT; #ab120633 abcam, Cambridge, UK) dissolved in Dimethyl sulfoxide (DMSO; #D8418 Sigma-Aldrich, St Louis, MO, USA) was used as a γ-secretase inhibitor and applied at 16 hours post-fertilization. DMSO alone was used as a control. A normal LED white light (GENTOS Co., Ltd., Tokyo, Japan) was used for photoirradiation experiments, and the photon flux density was measured (SEKONIC spectromaster C-7000, Tokyo, Japan) and adjusted to 150−200 µmol m$^{-2}$ s$^{-1}$. To create

dark conditions, the dishes containing larvae were simply wrapped with aluminum foil and kept in dark incubators until use.

### Single-cell RNA-seq
**[Data collection].** Embryos/larvae from a pair of sea urchins obtained in the Shimoda area at 24 hours (gastrula stage), 36 hours (prism stage), 48 hours (early pluteus stage), 72 hours (early 4-arm pluteus stage), and 96 hours (4-arm pluteus stage) were collected, dissociated, fixed, and dehydrated as followed by the method previously reported for sea urchin experiments[73]. Collected embryos/larvae were washed twice in ice-cold Ca$^{2+}$-free seawater including 1 mM EGTA. Then, they were dissociated in dissociation buffer (1.0 M glycine, 0.25 mM EDTA) for 10 min at 4 °C, with gentle pipetting every 2 min. Spin 500 g x 5 min and discard dissociation buffer, and cells are resuspended in Ca$^{2+}$-free seawater. Cell number and dissociation quality were measured at this time. After filtering by 20 µm mesh, cells are fixed in a final 80% ice-cold Methanol for 1 hour at 4 °C. Then, the samples were stored at −20 °C until use. The samples were rehydrated in 3x SSC containing 0.04% BSA, 1 mM DTT, and 0.2 U/µl RNase inhibitor. Single-cell libraries were prepared with 10x Genomics Chromium single-cell RNA-seq library preparation v3.1 to encapsulate single cells into droplets. Sequencing was performed with DNBSEQ ~ 350 M read/lane (sample) (Azenta Life Sciences, Tokyo, Japan).

**[Data pre-processing].** The reference genome file (HpulGenome_v1_scaffold.fa) and the annotation file (HpulGenome_v1.gff3) of *H. pulcherrimus* were retrieved from Hpbase. The GFF file was converted to the GTT file by gffread (v0.12.7)[74]. These files were used in Cell Ranger (v7.0.0). The index file of the STAR aligner[75] was generated by cellranger mkref and then used in cellranger cont, which counts the Unique Molecular Identifiers (UMIs) in each gene of each droplet. Cellranger count automatically removed the low-quality data derived from the empty droplet. Overall, the average cell number, reads per cell, and genes per cell are 7,580, 69,203, and 1,013, respectively. Data for each stage are presented in Supplementary Table 1. The Read10X function of Seurat (v4.3.0)[76] was then applied to the Cell Ranger output files for each batch, and the data were loaded in R (v.4.2.2) for subsequent data analysis.

**[Data integration].** To integrate 10 batch datasets into a single Seurat object, we followed the Seurat documentation (https://satijalab.org/seurat/articles/integration_introduction.html). We performed the standard analysis workflow, such as SCTransform, RunPCA, RunUMAP, FindNeighbors, and FindClusters in each batch, and then we performed the integration using SelectIntegrationFeatures, FindIntegrationAnchors, and IntegrateData functions.

**[Cell clustering and differential expression gene (DEG) detection].** After the integration, we performed the SCTransform[77] function to normalize/scale UMI counts and to extract highly variable genes. Principal Component Analysis (PCA) with the IRLBA algorithm[78] was performed with the RunPCA function, and the top 30 PC scores were used for RunUMAP to perform Uniform Manifold Approximation and Projection (UMAP)[79]. Also, using the top 30 PC scores, k-nearest neighbors (kNN) were constructed with $k = 20$ using the FindNeighbors function, and Louvain clustering[80] was performed on the kNN graph using the FindClusters function. Using the FindConservedMarkers function, the Wilcoxon rank-sum test[81] was performed between a cluster and all other clusters to detect the cluster-specific differential expression.

**[Data quality control (QC)].** To examine whether clusters are being formed due to the undesirable state of cells (e.g., cell health and presence of dead cells) and technical factors such as doublets, the following values were used to color each cell on the UMAP plot.

1. Number of detected genes
2. Total UMI counts
3. Percentage of UMI counts from the mitochondria-related genes
4. Percentage of UMI counts from the ribosomal-related genes
5. Estimated cell cycle by CellCycleScoring from Seurat
6. Doublet score by scDblFinder (v1.12.0)[82].

For 1. to 4., no cluster-specific bias was observed for these values. For 5. and 6., a slight bias was observed in some clusters, but it was indistinguishable from biological variation, such as the degree of stemness by cell type. Therefore, further cell filtering was not performed in this paper.

**[Cell-type annotation].** We check that the DEG list detected in each cluster contains cell type-specific markers based on the previous reports, and then convert cluster numbers to cell type labels. To check whether the integration by Seurat did not mislabel the same cluster number to different cell types, we also compared dot plots across all the batches and confirmed that the clusters sharing the same cluster number specifically express the same marker genes. Overall, we classified cells into 41 clusters and identified 16 major tissues, and focused on neurons in this study.

## Whole-mount in situ hybridization and immunohistochemistry

Whole-mount in situ hybridization was performed as described previously[38,83] with some modifications. cDNA mix from several embryonic stages was used to make RNA probes based on the *H. pulcherrimus* genome and transcriptome[84,85]. The samples were incubated with digoxygenin (Dig)-labeled RNA probes for opn5L (HPU_23194), tph (HPU_21307), foxQ2 (HPU_15608, HPU_15609), Z167(HPU_10494), ebf3 (HPU_03418), sna (HPU_14572), hmx (HPU_02863), six3 (HPU_00364), rx (HPU_04689), awh (HPU_17956), otx (HPU_04624), cry1 (HPU_09510), gcm (HPU_07306), lefty (HPU_15030), pura-like (HPU_23258), pde9a-like (HPU_03029), ets1/2 (HPU_15019), cnpy4 (HPU_05237), and pax1-9 (HPU_13853) at a final concentration of 0.4-1.2 ng/µl at 50 °C for 5 days. The Dig-labeled probes were detected with an anti-Dig POD-conjugated antibody (#11207733910 Roche, Basel, Switzerland) and treated with the Tyramide Signal Amplification System (#B40953, #D40955 TSA; ThermoFisher Scientific, Waltham, MA, USA, #NEL742001KT TSA Plus TMR; AKOYA Biosciences, Marlborough, MA, USA) for 8 min at room temperature (RT). For chromogenic detection, diluted anti-DIG antibody conjugated with alkaline phosphatase (#11093274910 Roche) (1:1000) was used for overnight incubation to detect the probe. After several washes in MOPS buffer, NBT/BCIP system (Promega Corporation, Madison, WI, USA) was used to detect the signal.

Whole-mount immunohistochemistry was also performed as described previously[38] with some modifications. The 3.7% formaline (in SW)-fixed samples were blocked with 1% skim milk in PBST for 1 hour at RT and incubated with primary antibodies (dilutions: mouse anti-Synaptotagmin B [SynB][86], rabbit anti-serotonin [#S5545; Sigma-Aldrich], 1:1000, rabbit anti-Nkx2.1[87]) overnight at 4 °C. The primary antibody was detected with a goat anti-mouse/rabbit IgG Alexa-Fluor-conjugated antibody (#A-31570, #A-21206, ThermoFisher Scientific) diluted 1:2000.

## Microinjection of morpholino anti-sense oligonucleotides (MO) and DNA

For microinjection, we used injection buffer (24% glycerol, 20 mM HEPES pH 8.0, and 120 mM KCl). The morpholino (Gene Tools, Philomath, OR, USA) sequences and the in-needle concentration with injection buffer were as follows:

Z167-MO1 (0.5 mM): 5′- GCACGTTTCTTTTTGTCATA −3′,
Z167-MO2 (0.5 mM): 5′- TGCATCCCCTTTTCTTTATA −3′,
Opn5L-MO1 (0.2 mM): 5′- TTTCCCTTTAATGCTCACTTTTC −3′,

Opn5L-MO2 (0.4 mM): 5′- CATTCGCATCCATCTTATCC −3′,
Sna-MO1 (1.0 mM): 5′- TTTTGACGAGAAAAGACCTCGGCAT −3′, and
Sna-MO2 (0.5-1.0 mM): 5′- GAATTATTCAAATCCGGTACAGCTC −3′,

Microinjections into fertilized eggs were performed as previously described[88]. After microinjection, the eggs were washed with FSW three times and stored with 50 µg/ml of kanamycin until the desired stages. The specificity of these morpholinos was confirmed in Fig. 2 and Supplementary Fig. 4 (Z167), Fig. 4 and Supplementary Fig. 6 (Sna), and Fig. 5 and Supplementary Fig. 7 (Opn5L).

The DNA construct for the putative cis-regulatory element of Z167 was prepared and injected as previously described[89] with some modification. Four thousand base-pairs of the genomic DNA of *H. pulcherrimus* were isolated and combined with a DNA sequence encoding Venus with Endo16 basal promoter[90].

## Microscopy and image analysis

The specimens were observed using a fluorescence microscope (IX73, Evident, Tokyo, Japan) and a confocal laser scanning microscope (FV10i, Evident). Live-imaging using a DNA construct was captured using Andor BC43 (Oxford Instruments, Abington, Oxfordshire, UK). All transmission images were taken with the IX73 and the digital camera (DP74, Evident). Panels and drawings for the figures were made using Fiji, Adobe Photoshop, and Microsoft PowerPoint. A part of Fig. 5 is drawn in Biorender under the license for publication (agreement # ME2803VZCQ).

## Analysis of the molecular properties of the Opn5L protein

To analyze the spectral property of sea urchin Opn5L proteins, their recombinant proteins were obtained in cultured cells according to the previous report about vertebrate Opsin5[15]. To improve the expression yield of the recombinant proteins, 187 and 188 amino acid residues were truncated from the C-terminus of cDNAs of *H. pulcherrimus* (HPU_23194) and *S. purpuratus* (SPU_019120) Opn5L, respectively. These cDNAs were tagged with the epitope sequence of the anti-bovine rhodopsin monoclonal antibody Rho1D4 at the C-terminus and were introduced into the mammalian expression vector pCAGGS. The plasmid DNA was transfected into HEK293S cells using the calcium phosphate method. Five µM 11-*cis* retinal was added to the medium 24 hours after transfection, and the cells were kept in the dark before the collection of the cells 48 hours after transfection. The reconstituted pigments were extracted from cell membranes with 1 % dodecyl maltoside (DDM) in Buffer A (50 mM HEPES, 140 mM NaCl, pH 6.5) and were purified using Rho1D4-conjugated agarose. The purified pigments were eluted with 0.02 % DDM in Buffer A containing the synthetic peptide that corresponds to the C-terminus of bovine rhodopsin. All the procedures were carried out on ice under dim red light. UV/Vis absorption spectra were recorded using a Shimadzu UV2450 or UV2600 spectrophotometer at 0 °C. The sample was irradiated with light which was generated by a 1-kW tungsten halogen lamp (Master HILUX-HR, Rikagaku Seiki) and passed through optical filters (Y-52 or UV-D36C, AGC Techno Glass).

The elevation of $Ca^{2+}$ level by Opn5L in HEK293S cells was measured using an aequorin-based bioluminescence assay[28,91]. HEK293S cells were seeded in 96-well plates at a density of 60,000 cells/well in medium (D-MEM/F12 containing 10 % FBS). After overnight incubation, the plasmid DNA was transfected into the cells (100 ng/well) using the polyethyleneimine transfection method. After incubation for 6 hr, 11-*cis* retinal was added to the medium (final concentration: 5 µM). After overnight incubation, the medium was replaced with the equilibration medium ($CO_2$-independent medium containing coelenterazine h and 10 % FBS). After incubation for 2 hr at room temperature, luminescence from the cells was measured using a microplate reader (SpectraMax L, Molecular Devices). The cells were

irradiated for 5 sec with UV LED light (365 nm) to trigger the change of the luminescence.

## Statistical analysis

No statistical methods were used to predetermine the sample sizes. All n numbers are described in the figure legends. To compare the two groups, we used Welch's t-test (two-tailed) with a significance level of 0.001, 0.01 or 0.05. For Fig. 5c: [control dark vs light] $t$-value = 7.8846, degrees of freedom [d.f.] = 4.625, $p$-value = 0.0007504, [Opn5L-MO1 dark vs light] $t$-value = 1.3429, d.f. = 4.1057, $p$-value = 0.2487. For Supplementary Fig. 7b: [control dark vs light] t-value = 16.163, d.f. = 4.1718, $p$-value = 6.401e-05, [Opn5-MO2 dark vs light] t-value = 3.0756, d.f. = 7.6589, $p$-value = 0.01604, [control light vs Opn5-MO2 light] $t$-value = 4.6669, d.f. = 6.9025, $p$-value = 0.002383. To compare more than two groups, we used one-way ANOVA followed by Tukey's post hoc test with a significance level of 0.01 or 0.05, and the following F values (F) and d.f. For Fig. 4f: F = 30.8021, d.f. = 3, [36 h vs 42 h], $p$-value = 0.8565991, [36 h vs 48 h], $p$-value = 0.009274, [36 h vs 54 h], $p$-value = 0.0010053.

## Reporting summary

Further information on research design is available in the Nature Portfolio Reporting Summary linked to this article.

## Data availability

Source data for all Figures and Supplementary Figs. are provided with the paper. Sequence data can be found in the genome database of *Hemicentrotus pulcherrimus*, HpBase (http://cell-innovation.nig.ac.jp/Hpul/)[84]. Other data are available from the corresponding author upon request. The raw and processed next generation sequencing data have been deposited in NCBI's Gene Expression Omnibus and are accessible through GEO Series accession number GSE265747. Source data are provided with this paper.

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

## Acknowledgements

We thank Y. Nakajima, R.D. Burke, and H. Katow for essential reagents. We thank M. Kiyomoto, M. Ooue, J. Takano, G. Northen, H. Abe, M. Washio, M. Yamaguchi, and JF Izu/Shimoda for collecting and keeping the adult sea urchins. We thank Prof. Robert S. Molday for the generous gift of a Rho1D4-producing hybridoma, and Prof. Jeremy Nathans for providing the HEK293S cell line. This work is supported by JST PRESTO Grant number JPMJPR194C, JST A-STEP Grant number JPMJTR204E, JSPS KAKENHI Grant number 23K23933, the Toray Science Foundation, and Takeda Science Foundation to S. Yaguchi, JST PRESTO Grant number JPMJPR1945 and JSPS KAKENHI Grant Numbers 19K20406 and 23K11312 to K.Tsuyuzaki, AMED CREST Grant number 22gm1510007, and The Naito Foundation to T. Yamashita.

## Author contributions

Studies were designed by J.Y., K.T., and S.Y. Preliminary data were acquired by J.Y., K.T., and S.Y. Sea urchin data were acquired by J.Y., K.T., N.S., T.Y., and S.Y. Culture cell data were acquired by I.S., A.H., and T.Y. The manuscript was written by J.Y., K.T., T.Y., T.Y., and S.Y. with input from all authors.

## Competing interests

The authors declare no competing interests.
