## [Peer Review file · Nature Communications]

Shared Evolutionary Origin of Non-Visual Photoreceptive Brain Specification

Corresponding Author: Professor Shunsuke Yaguchi

Version 0:

Reviewer comments:

Reviewer #1

(Remarks to the Author)

This manuscript by Yaguchi et al describes a previously uncharacterized population of serotonergic neurons located in the posterior apical organ in sea urchins. The authors use a combination of single-cell transcriptomics, expression, functional and behavioral assays to show that these cells are non-visual photoreceptors that are essential for light perception in sea urchins. Overall, the authors want to make the case that because these cells have a similar gene repertoire to non-visual photoreceptors located in the vertebrate diencephalon that this indicates that posterior non-visual neurons emerged in the common deuterostome ancestor. Most of the evidence supports their idea. However, there are several issues that the authors should consider addressing to strengthen the study.

Major concerns

In the abstract the authors should make it clear that this is not the first study to challenge the idea the sea urchin is "brain-less". Their lab and others have provided evidence in several other studies challenging this notion. This study provides additional evidence to strengthen their argument.

Introduction - The overall idea is to convince the reader that the population on posterior serotonergic cells provides evidence to challenge the idea that the apical sensory organ is brain-like. Therefore, the authors should consider adding more context to the introduction. Describing previous studies that already proposed this idea. These should probably include recent studies showing the conservation of the apical sea urchin GRNs among many metazoans including invertebrate and vertebrate chordates where many components are expressed in the forebrain/hypothalamus.

Lines 105-109. A better explanation of why they used DAPT to block Delta-Notch signaling to perform their single-cell seq transcriptomics. This may not be clear to many readers. Also, they should make it clear how they identified two distinct neural clusters.

Throughout the manuscript and figures the authors should consider using embryonic stages along with the hours post-fertilization in parentheses. Since the timing of urchin development varies with each model it makes it difficult for even sea urchin biologists to know which stage is being referenced. This is even more esoteric for developmental biologists that are not familiar with the sea urchin development.

On lines 131-132 the authors state that the dorsal neurons are expressing opn5L but do not show these data in figure 1, just a diagram (and they don't indicate that the data are in extended Fig. 3). The researchers should add an additional panel in fig. 1 showing that opn5L is in the posterior neurons as well as a side view that clearly shows it is there like in Fig. 1p.

In figure 2a-h in situ for several candidate dorsal neural genes are shown and they say that these are clearly dorsal. But they do not show data to support this statement in the panel (or in the extended figures except for opn5l). They do show that z167 appears to be in the posterior neurons in Fig. 2i-s. The authors should consider leading with the z167 data in figure 2 (including the functional work) then move to the other dorsal neural genes they have in fig. 2 a-h. But then they need to show double in situ with either zic127 or opn5l to show these genes are expressed in dorsal neurons. Otherwise, their

conclusions are unsupported by the data. Also, maybe consider making the panel 2a-h into a separate fig. 3 since it will get bigger with the double in situ data (if there is room)?

The researchers use two different non-overlapping morpholinos for each of their experiments but do not show the data showing that both morpholinos worked. Which of the morpholino pairs did the authors use for the data on the figures? Extended figures should be included showing the phenotypes of the morpholino not used in the main figures. Also, while not ideal at later stages the authors should try and rescue the morpholino phenotypes to alleviate concerns for off target effects.

Minor concerns

In the abstract I'm concerned that readers will not necessarily understand what the term "Delta/Notch interference-based single-cell transcriptomics" means. Is there a better way to state what you used to find the genes of interest? Maybe "we used single-cell transcriptomics and in situ hybridization to identify candidate genes....." and then mention that they used DAPT to increase the single-cell seq copy numbers or something?

Line 54 of abstract: Complexification is not a good word choice here since it is only used when describing vectors in mathematics.

Also, in the abstract they mention posterior serotonergic neurons kind of out of the blue. These have not been described before so most readers will not know why this is interesting. The authors should consider briefly adding more context.

Line 79 looks like it contains a typo or a placeholder that was missed in the editing process with a sentence ending in "..and etc.". The authors should complete the sentence.

Lines 140-142. DAPT-treated embryos are again mentioned but with no context.

In figure 2 the researchers use the markers for the Tryptophan Hydroxylase and FoxQ2 genes but never explain why so they should consider doing so.

Reviewer #2

(Remarks to the Author)

This study investigates the expression localization of non-visual opsin and associated specifier genes of brain in sea urchin larvae, and indicate that the brain regions of sea urchin larvae and vertebrates may share a common ancestral origin. The results of work challenge the current evolutionary trajectory of the non-visual opsin in deuterostomes, and break the hypothesis that sea urchins have no brain architecture. These data have a positive impact on analyzing the mechanism and evolution of non-visual photoreceptive brain structures in deuterostomes. However, explanations for some results that are contradictory to previous studies, and discussions of other additional unintended outcomes under target treatment are still ambiguous and needed to be discussed.

Main:

- 1) What is the specific expression distribution of z167 in anterior/dorsal serotonergic neurons generated from scRNA-seq data? Does it approximate or consistent with the results of in situ hybridization?
- 2) The qualification of the spatial relationship between dorsal and anterior serotonergic neurons was performed by counting the cell numbers between those two cell types. However, the histogram showing the changes in cell count over time remains relatively abstract, and the huge cell number difference between 36hr and 54hr is still relatively obvious. If the manuscript can present the in situ visualization image of the cell numbers under the measurement time nodes (36, 42, 48, 54hr) like Figure 3f, it will help readers understand the spatial relationship of the two cell types.
- 3) What is the expression pattern of z167 in Sna-MO2 group? Is it similar to Sna-MO1 that causes z167 down-regulation? The relationship between Sna silencing and z167 down-regulation is still ambiguous in migration of neurons, and I hope the author can further elaborate. Moreover, under the condition that the migration of PMCs (primary mesenchyme cells) is slowed down by Sna silencing, how to ensure the synchronization of measurement nodes in control and Sna-MO1/2, and what is the basis for node determination?
- 4) In comparison of swimming rate between control and Opn5L-MO2 morphants, although the results are logical and lead to straightforward conclusions, large degree of data dispersion in Opn5L-MO2 morphants due to a low n numbers may not be sufficiently persuasive.

Minor/Miscellaneous points:

- 1) Line 79, the spelling of "paraventricular" should be "paraventricular".
- 2) Figure 2q&2r, the expression patterns of tph signals in dorsal serotonergic neurons at 42h and 48h are ambiguous for readers. The strong signal presented by z167 overlapped with the tph signal at those two time nodes. A redrawn

representation like 36h may make it easier for the reader to understand the localization and migration of tph at those two time nodes in dorsal serotonergic neurons.

3) Line 856, Figure 2v-2v" are not demonstrated in the manuscript, they might be u-u"?

Reviewer #3

(Remarks to the Author)

In the paper "Shared evolutionary origin of non-visual photoreceptive brain specification", the authors describe their discovery of the expression of non-visual opsin, *Opn5*, in some serotonergic neurons of the anterior neurectoderm of the sea urchin *Hemicentrotus pulcherrimus*.

They first performed single-cell mRNA sequencing in wild-type and DAPT-treated larvae at different stages of development. They identified two clusters of serotonergic neurons, one of which expressed *Opn5L*, which they verified in situ. They identified that these neurons that express *Opn5L* are the dorsal serotonergic neurons. By performing differential gene expression analysis between *Opn5L*-positive and negative neurons, they identified a series of genes that are expressed specifically in the *Opn5L*-positive neurons that they verified experimentally that they are expressed dorsally. Then, the authors argue that these dorsal serotonergic neurons migrate towards the anterior region. Finally, they show that *Opn5L* responds to UV light and regulates sinking behavior.

While I think that this paper has some interesting observations, the claims that are made along the way are exaggerated to a great extent. I mention some examples:

a) In the abstract, the authors state "This is partly due to the assumption that echinoderms, a key group within this clade, are recognized as brain-less^{1,3,4}. Here, we challenge this assumption by demonstrating that sea urchin larvae express non-visual opsin in the posterior region of their "brain"". First of all, references 1, 3, 4 do not recognize echinoderms as brain-less. Second, the fact that there are serotonergic neurons expressing non-visual opsins would not suffice to challenge this assumption.

b) In the end of the Introduction, they state "our findings support the hypothesis that the origins of the brain observed in chordates may have been established in the common ancestor of echinoderms and chordates." Of course, the origins of the chordate brain would have been established in the common ancestor. What is the alternative?

c) "Our findings reveal a previously unreported migratory behavior". The findings presented in this paper include the expression of *Snail* and the quantification of cells between anterior and dorsal serotonergic neurons. This does not suffice to describe a migratory behavior.

d) Same later "our data provide compelling evidence that sea urchin embryos exhibit migratory behavior"

e) In the discussion "Our findings challenge the long-standing notion that echinoderms, as brain-less deuterostomes, lack centralized neural structures comparable to those in chordates^{2,13,60}" Again, I strongly disagree that the data presented in this paper challenge this notion. They don't show the presence of a centralized neural structure comparable to those in chordates.

In general, I think that the paper falls short to address the question that it sets out to explore (based on the abstract and the introduction). I believe that the authors should re-evaluate the data and present them in a more realistic context.

On top of the above, I have also other comments that could improve the paper, in my opinion:

- Introduction: there is not sufficient information about what is known about non-visual opsins in the echinoderm brain.
- Lines 107-109: the authors should comment on the impact of DAPT in the neurons. While their interpretation will not change, as they only present control data, it is a useful piece of information for the reader.
- Line 113: In Figures 1c-f, it would be good to specify if the larvae are wild-type or DAPT-treated.
- Line 140: How were the "regions" compared? Do the authors mean cells? Or clusters? Please be more precise.
- Line 164-166: The authors claim that *rx* is expressed in *Opn5L*-expressing cells and reference Figure 2h, but in this Figure *Opn5L* expression is not shown.
- In Figure 2, in general, the authors should present double stains with *tph* and/or *opn5L*.
- Lines 204: There is no reference for this "lonstanding view"
- Line 300: What do the authors mean by "this study"? This study doesn't show this.
- Reference 3 is incorrect. The title of the paper has changed in Science Advances.
- Video S1 is not really supportive of migration. Same for Figure 3e-e'.
- For Table 1, very minimal information is provided as to how these genes were identified and how the genes to be tested were selected.

Version 1:

Reviewer comments:

Reviewer #1

(Remarks to the Author)

The authors have addressed all of my concerns and I now recommend this study for publication.

Reviewer #2

(Remarks to the Author)

All the reasonable questions posed by the reviewers have been answered comprehensively. I have no additional suggestions to offer.

Reviewer #3

(Remarks to the Author)

The authors have addressed the reviewers' comments successfully and, in my opinion, this led to the improvement of the manuscript. A couple concerns remain:

- Based on Supplementary Figure S2, Opn5L is also expressed in other cells besides the dorsal serotonergic neurons. How do the authors know that the effect of the Opn5L knock-down is because of its absence in the dorsal serotonergic neurons and not in the other cells? On that note, it would be good to highlight what kind of cells these are.
- I am still not convinced that Figure 4 shows migration. While the movie is much better now and indeed shows movement, migration is an active, energy-dependent process. I am even more concerned as they use this movement to report "the first neuronal migration in Ambulacraria reshaping our understanding of deuterostome nervous system evolution". I still believe that with the current data, no neuronal migration has been shown.

Minor comments:

- I think that the phrase "and enhanced detection of rare neuronal populations by inhibiting Delta-Notch signaling with DAPT" in the Abstract is an unnecessary detail and can be confusing for the reader.
- Line 163: "expressing" should probably be "expression in"
- In Figures 2k,l, why don't the authors co-stain with ebf3 to show beyond doubt that these are the anterior serotonergic neurons?
- Line 223: "functioned" should be "function" or "functional"
- Line 275: "overlapped" should be "overlapping"

Reviewer #1 (Remarks to the Author):

This manuscript by Yaguchi et al describes a previously uncharacterized population of serotonergic neurons located in the posterior apical organ in sea urchins. The authors use a combination of single-cell transcriptomics, expression, functional and behavioral assays to show that these cells are non-visual photoreceptors that are essential for light perception in sea urchins. Overall, the authors want to make the case that because these cells have a similar gene repertoire to non-visual photoreceptors located in the vertebrate diencephalon that this indicates that posterior non-visual neurons emerged in the common deuterostome ancestor.

Most of the evidence supports their idea. However, these are several issues that the authors should consider addressing to strengthen the study.

Major concerns

In the abstract the authors should make it clear that this is not the first study to challenge the idea the sea urchin is "brain-less". Their lab and others have provided evidence in several other studies challenging this notion. This study provides additional evidence to strengthen their argument.

We agree with the reviewer. This point has also been raised by other reviewers, so we have revised the text to clarify that, although echinoderms have long been considered to lack a brain-like structure, recent discussions have emerged suggesting that they may possess brain-like features based on gene expression profiles and functional evidence. Due to the word limit of Nature Communications, we have substantially and concisely revised the abstract.

Introduction - The overall idea is to convince the reader that the population on posterior serotonergic cells provides evidence to challenge the idea that the apical sensory organ is brain-like. Therefore, the authors should consider adding more context to the introduction. Describing previous studies that already proposed this idea. These should probably include recent studies showing the conservation of the apical sea urchin GRNs among many metazoans including invertebrate and vertebrate chordates where many components are expressed in the forebrain/hypothalamus.

We agree with the reviewer's comment. We have made substantial revisions to the end of the Introduction. As suggested, we now refer to studies that indicate the presence of brain-like features in echinoderms, including those that highlight hypothalamus-like characteristics comparable to vertebrates, and have cited them appropriately.

Lines 105-109. A better explanation of why they used DAPT to block Delta-Notch signaling to perform their single-cell seq transcriptomics. This may not be clear to many readers. Also, they should make it clear how they identified two distinct neural clusters.

Thank you for this comment. We agree with the reviewer that our initial explanation for using DAPT in the scRNA-seq analysis was insufficient. We have now added a detailed rationale for this choice. In addition, we have included an explanation of how we identified the two clusters of serotonergic neurons in the UMAP.

Throughout the manuscript and figures the authors should consider using embryonic stages along with the hours post-fertilization in parentheses. Since the timing of urchin development varies with each model it makes it difficult for even sea urchin biologists to know which stage is being referenced. This is even more esoteric for developmental biologists that are not familiar with the sea urchin development.

Thank you for this comment. As suggested, we have added the developmental stages

to the images and corresponding text. To avoid cluttering the figures, the stage is indicated only at the first appearance within each figure.

On lines 131-132 the authors state that the dorsal neurons are expressing opn5L but do not show these data in figure 1, just a diagram (and they don't indicate that the data are in extended Fig. 3). The researchers should add an additional panel in fig. 1 showing that opn5L is in the posterior neurons as well as a side view that clearly shows it is there like in Fig. 1p.

Thank you for this. Yes, it is true that we did not show *opn5L in situ* image in the main figure as illustrated. We added the *in situ* hybridization image in Figure 1h, h' (lateral view).

In figure 2a-h in situs for several candidate dorsal neural genes are shown and they say that these are clearly dorsal. But they do not show data to support this statement in the panel (or in the extended figures except for opn5l). They do show that z167 appears to be in the posterior neurons in Fig. 2i-s. The authors should consider leading with the z167 data in figure 2 (including the functional work) then move to the other dorsal neural genes they have in fig.2 a-h. But then they need to show double in situs with either zic127 or opn5l to show these genes are expressed in dorsal neurons. Otherwise, their conclusions are unsupported by the data. Also, maybe consider making the panel 2a-h into a separate fig. 3 since it will get bigger with the double in situ data (if there is room)?

Thank you for this comment. We agree with the reviewer's point. We first presented the double *in situ* hybridization data for *opn5L* and *z167*, and then introduced the *z167*-related findings and corresponding functional data in Figure 2. In Figure 3, we separately provided double *in situ* hybridization data showing the expression of dorsal neural genes together with *z167*, in order to demonstrate that these genes are expressed in dorsal serotonergic neurons. The corresponding text has been revised accordingly to reflect these changes accurately.

The researchers use two different non-overlapping morpholinos for each of their experiments but do not show the data showing that both morpholinos worked. Which of the morpholino pairs did the authors use for the data in the figures? Extended figures should be included showing the phenotypes of the morpholino not used in the main figures. Also, while not ideal at later stages the authors should try and rescue the morpholino phenotypes to alleviate concerns for off target effects.

Thank you for this comment. We indeed used a second, non-overlapping morpholino in all experiments involving *Z167*, *Sna*, and *Opn5L*, and the corresponding data are included. However, we now realize that this was not clearly stated in the main text, which may have prevented readers from recognizing this point. In the revised version, we have explicitly indicated the use and location of the second morpholino in the text. Regarding the reviewer's suggestion of rescue experiments, while we agree that this approach is generally ideal, we chose not to pursue it in this case. As the reviewer also notes, overexpression in inappropriate cell types may lead to non-specific effects that do not reflect physiological relevance. Therefore, we decided not to perform rescue experiments in this study.

Minor concerns

In the abstract I'm concerned that readers will not necessarily understand what the term "Delta/Notch interference-

based single-cell transcriptomics” means. Is there a better way to state what you used to find the genes of interest? Maybe “we used single-cell transcriptomics and in situ hybridization to identify candidate genes.....” and then mention that they used DAPT to increase the single-cell seq copy numbers or something?

Thank you for this. We fixed it. Due to the word limit of Nature Communications, we have substantially and concisely revised the abstract.

Line 54 of abstract: Complexification is not a good word choice here since it is only used when describing vectors in mathematics.

We deleted it in the course of substantially revising the abstract.

Also, in the abstract they mention posterior serotonergic neurons kind of out of the blue. These have not been described before so most readers will not know why this is interesting. The authors should consider briefly adding more context.

We agree the reviewer's comment. Due to the word limit in the abstract, we did not have sufficient space to explain the posterior serotonergic neurons in detail. Therefore, we referred to them simply as serotonergic neurons of the neuroectoderm. Further details are provided in the Introduction and Results sections.

Line 79 looks like it contains a typo or a placeholder that was missed in the editing process with a sentence ending in “..and etc.”. The authors should complete the sentence.

Thank you for this comment. We modified it to “...such as the diencephalon, and later develop into the hypothalamus and the paraventricular organ.”

Lines 140-142. DAPT-treated embryos are again mentioned but with no context.

Thank you for this comment. We added the explanation why we mentioned about DAPT-treated embryo after the sentence. “Since DAPT treatment increases the number of neurons, it enhances the visibility of genes that are otherwise rare in control embryos. Taking advantage of this property, we performed in situ hybridization and successfully detected the expression of these genes.”

In figure 2 the researchers use the markers for the Tryptophan Hydroxylase and FoxQ2 genes but never explain why so they should consider doing so.

We did not provide an explanation for *tph* here, as it was already described in Figure 1, and repeating it would have been redundant. In contrast, as the reviewer correctly pointed out, we had not explained *foxQ2*, despite using it in Figure 1. To address this, we have added a brief explanation at its first appearance in Figure 1: “FoxQ2 is an initial specifier of the anterior neuroectoderm. It is broadly expressed throughout the ectoderm at early stages but becomes restricted to the anterior-most region of the neuroectoderm as development progresses.”. We also explain FoxQ2 at Figure 2 part, “The anterior neuroectoderm specifier FoxQ2 is localized to the most anterior region by 36 hours, and it is known that serotonergic neurons differentiate at its dorsal boundary. However, *z167*-expressing cells at 36 hours are not adjacent to the *foxQ2* domain; instead, they differentiate in a distinct region located more dorsally than the *foxQ2* area. In contrast, cells expressing *ebf3* are positioned adjacent to the dorsal boundary of the *foxQ2* domain.”

Reviewer #2 (Remarks to the Author):

This study investigates the expression localization of non-visual opsin and associated specifier genes of brain in sea urchin larvae, and indicate that the brain regions of sea urchin larvae and vertebrates may share a common ancestral origin. The results of work challenge the current evolutionary trajectory of the non-visual opsin in deuterostomes, and break the hypothesis that sea urchins have no brain architecture. These data have a positive impact on analyzing the mechanism and evolution of non-visual photoreceptive brain structures in deuterostomes. However, explanations for some results that are contradictory to previous studies, and discussions of other additional unintended outcomes under target treatment are still ambiguous and needed to be discussed.

Main:

1) *What is the specific expression distribution of z167 in anterior/dorsal serotonergic neurons generated from scRNA-seq data? Does it approximate or consistent with the results of in situ hybridization?*

The results of scRNA-seq and z167 in situ hybridization are consistent. However, as pointed out by Reviewer #2, the presentation was not sufficiently clear, and as also noted by Reviewer #1, we have now moved the detailed analysis of z167 to the beginning of Figure 2. In this revision, we added the following explanation: "The UMAP in Fig. 2i represents the same region as shown in Fig. 1d and e. Both the light-blue dashed and yellow solid regions express tph, but they are distinct in gene expression: the light-blue dashed region is *ebf3⁺/opn5L⁻/z167⁻*, while the yellow solid region is *ebf3⁻/opn5L⁺/z167⁺*. This mutually exclusive gene expression pattern is supported by the *in situ* hybridization results in Figure 2."

2) *The qualification of the spatial relationship between dorsal and anterior serotonergic neurons was performed by counting the cell numbers between those two cell types. However, the histogram showing the changes in cell count over time remains relatively abstract, and the huge cell number difference between 36hr and 54hr is still relatively obvious. If the manuscript can present the in situ visualization image of the cell numbers under the measurement time nodes (36, 42, 48, 54hr) like Figure 3f, it will help readers understand the spatial relationship of the two cell types.*

Thank you for this comment. We agree that the explanation was unclear, as the reviewer pointed out. The data presented in this graph were obtained by counting cells from multiple microscopy samples used for the *in situ* hybridization experiments shown in Figure 2. We have now clarified this point in the revised text. "To quantify this migration, we measured the distance, i.e. counted the number of cells between *tph*-positive/z167-negative anterior serotonergic neurons and z167-positive dorsal serotonergic neurons from the samples we showed in Figure 2j-n and confirmed that the distance was reduced (Fig. 4f, g)."

3) *What is the expression pattern of z167 in Sna-MO2 group? Is it similar to Sna-MO1 that causes z167 down-regulation? The relationship between Sna silence and z167 down-regulation is still ambiguous in migration of neurons, and I hope the author can further elaborate. Moreover, under the condition that the migration of PMCs (primary mesenchyme cells) is slowed down by Sna silencing, how to ensure the synchronization of measurement nodes in control and Sna-MO1/2, and what is the basis for node determination?*

Thank you for this comment. As reported in previous studies, Sna-MO1 showed stronger effects, so we initially presented only the data for Sna-MO1. However, since Sna-MO2 yielded comparable results, we have now included those data in Extended Data Fig. 6. The corresponding figure legend has also been revised accordingly.

Based on the presence of a tripartite gut, an open mouth, and elongation of skeletal rods in the pluteus configuration, we judged that the morphants had eventually reached the pluteus larval stage, albeit with some developmental delay, and proceeded with the following observations.

4) *In comparison of swimming rate between control and Opn5L-MO2 morphants, although the results are logical and lead to straightforward conclusions, large degree of data dispersion in Opn5L-MO2 morphants due to a low n numbers may not be sufficiently persuasive.*

Thank you for this comment, and we fully agree. As the reviewer pointed out, the original analysis was based on only three batches, so we increased the number to five in the final version. Details, including exact n numbers, are provided in the legend of Extended Data Fig. 7. Although Opn5L-MO2 appears slightly less effective than Opn5L-MO1, both show the same overall trend, and thus our conclusion that Opn5L is involved in light-dependent regulation of swimming behavior remains unchanged.

i

2) *Figure 2q & 2r, the expression patterns of tph signals in dorsal serotonergic neurons at 42h and 48h are ambiguous for readers. The strong signal presented by z167 overlapped with the tph signal at those two time nodes. A redrawn representation like 36h may make it easier for the reader to understand the localization and migration of tph at those two time nodes in dorsal serotonergic neurons.*

Thank you for this comment. We agree with the suggestion. Since the 42 hr double *in situ* hybridization image is not substantially different from those at 36 and 48 hr, we removed the 42 hr panel to improve the overall balance of the figure. Instead, we added a 48 hr *tph*-only image (Figure 2l, m in the revised version).

3) *Line 856, Figure 2v–2v" are not demonstrated in the manuscript, they might be u–u"?*

Thank you for this. We fixed it in the revised version.

Reviewer #3 (Remarks to the Author):

*In the paper "Shared evolutionary origin of non-visual photoreceptive brain specification", the authors describe their discovery of the expression of non-visual opsin, Opn5, in some serotonergic neurons of the anterior neurectoderm of the sea urchin *Hemicentrotus pulcherrimus*. They first performed single-cell mRNA sequencing in wild-type and DAPT-treated larvae at different stages of development. They identified two clusters of serotonergic neurons, one of which expressed Opn5L, which they verified *in situ*. They identified that these neurons that express Opn5L are the dorsal serotonergic neurons. By performing differential gene expression analysis between Opn5L-positive and negative neurons, they identified a series of genes that are expressed specifically in the Opn5L-positive neurons that they verified*

experimentally that they are expressed dorsally.

Then, the authors argue that these dorsal serotonergic neurons migrate towards the anterior region. Finally, they show that Opn5L responds to UV light and regulates sinking behavior. While I think that this paper has some interesting observations, the claims that are made along the way are exaggerated to a great extent. I mention some examples:

a) In the abstract, the authors state “This is partly due to the assumption that echinoderms, a key group within this clade, are recognized as brain-less^{1,3,4}. Here, we challenge this assumption by demonstrating that sea urchin larvae express non-visual opsin in the posterior region of their ‘brain’”. First of all, references 1, 3, 4 do not recognize echinoderms as brain-less. Second, the fact that there are serotonergic neurons expressing non-visual opsins would not suffice to challenge this assumption.

We thank the reviewer for pointing this out. We agree that the references cited in our original sentence do not explicitly refer to echinoderms as “brain-less,” and we acknowledge that such a characterization may not be appropriate as a general description of the current literature. We have therefore revised the sentence to reflect that the presence or absence of a brain-like structure in echinoderm larvae remains a topic of discussion, rather than framing it as a broadly accepted assumption.

*Due to the word limit of Nature Communications, we have substantially and concisely revised the abstract.

We also appreciate the opportunity to clarify that our argument does not rely solely on the presence of serotonergic neurons expressing non-visual opsins. Rather, our findings build on previous work, including from our own group, showing that these neurons are involved in coordinating larval behavior in response to external stimuli. We believe that the integration of molecular, anatomical, and behavioral evidence contributes incrementally to defining the nature and function of neural centers in echinoderm larvae, and may help inform the broader discussion of brain evolution in deuterostomes.

b) In the end of the Introduction, they state “our findings support the hypothesis that the origins of the brain observed in chordates may have been established in the common ancestor of echinoderms and chordates.” Of course, the origins of the chordate brain would have been established in the common ancestor. What is the alternative?

We thank the reviewer for raising this point. We agree that the statement in the original version may have implied an overly broad conclusion. In the revised manuscript, we have modified the sentence to more accurately reflect the nature of our contribution. Rather than asserting a definitive evolutionary origin, we now emphasize that our findings add support to an ongoing hypothesis that certain aspects of brain-related organization may have emerged in the common ancestor of chordates and echinoderms.

We hope this revision appropriately addresses the reviewer’s concern while maintaining the broader relevance of our data to ongoing discussions in evolutionary neurobiology.

c) “Our findings reveal a previously unreported migratory behavior”. The findings presented in this paper include the expression of Snail and the quantification of cells between anterior and dorsal serotonergic neurons. This does not suffice to describe a migratory behavior.

d) Same later “our data provide compelling evidence that sea urchin embryos exhibit migratory behavior”

We treat these two comments and the forthcoming one about the movie as related, and respond to them collectively.

Thank you for this comment. We agree that the movie presented in the original version was difficult to interpret and did not provide convincing evidence for readers. To address this, we revised the movie to cover a longer time window and a wider field of view, allowing for a clearer visualization of the relevant cellular events. We believe that, when viewed together with the movie, the supporting cell count data will help convince readers that the observed cells are indeed undergoing positional changes.

In response to the reviewer's concern, we also agree that the sentence beginning with "Our findings reveal..." and the use of the word "compelling" were overstated. We have removed these expressions accordingly. In particular, we judged that the "Our findings reveal..." sentence was inappropriate both in content and in placement, and therefore deleted it from the revised version.

e) In the discussion "Our findings challenge the long-standing notion that echinoderms, as brain-less deuterostomes, lack centralized neural structures comparable to those in chordates^{2,13,60}" Again, I strongly disagree that the data presented in this paper challenge this notion. They don't show the presence of a centralized neural structure comparable to those in chordates.

We thank the reviewer for their important comment. We acknowledge that our data do not demonstrate the presence of a centralized neural structure equivalent to those found in chordates. To better reflect the scope of our findings, we have revised the discussion to clarify that our aim is not to assert anatomical equivalence, but rather to propose that sea urchins may possess a neural domain that functionally integrates environmental cues to regulate behavior. This perspective contributes to ongoing discussions about the diversity of neural organization across deuterostomes.

In general, I think that the paper falls short to address the question that it sets out to explore (based on the abstract and the introduction). I believe that the authors should re-evaluate the data and present them in a more realistic context.

We appreciate the reviewer's thoughtful and constructive feedback. We agree that, although there has been long-standing debate on whether echinoderms possess a brain, definitive molecular or functional evidence has remained limited.

Our group has previously contributed to this topic by demonstrating functional aspects of neural circuits in echinoderm larvae, particularly their capacity to integrate environmental stimuli into coordinated behavior. Together with recent gene expression studies from our lab and others, these findings have begun to suggest that brain-like properties may indeed exist in echinoderm larvae.

In this study, we aimed to expand on this view by characterizing a population of non-visual opsin-expressing neurons—key molecular players in brain-associated light sensing—in the apical region of sea urchin larvae. We acknowledge that in our effort to underscore the significance of these findings, some statements may have been overstated. In response to the reviewer's comment, we have revised the manuscript throughout to present our conclusions in a more balanced and realistic manner.

Specifically, we now emphasize three major findings:

- (1) Non-visual opsin (Opn5L) is expressed in a subset of apical serotonergic neurons;
- (2) These neurons exhibit a gene expression signature overlapping with that of the

chordate forebrain/diencephalon; and

(3) They functionally mediate light-dependent behavioral changes, suggesting their role in coordinating external cues and behavior.

Importantly, we also note that non-visual opsins are likely to predate the evolution of image-forming eyes, especially within the deuterostome lineage, where the acquisition of complex visual systems appears to be a later event. Therefore, non-visual opsins may represent a more ancestral feature of photoreceptive brain structures, and their presence in echinoderm larvae offers a unique opportunity to explore early neural circuits that integrate environmental light cues.

Rather than claiming the presence of a structurally centralized brain, we now describe this region as a brain-like with both transcriptional and functional features that may provide valuable insight into the evolutionary origins of the deuterostome brain.

On top of the above, I have also other comments that could improve the paper, in my opinion:

Introduction: there is not sufficient information about what is known about non-visual opsins in the echinoderm brain.

Thank you for this comment. We agree that it is important to discuss opsins in echinoderms. In the revised version, we have added a brief overview of echinoderm opsins, particularly those in sea urchins, as they have been subject to functional analysis.

Lines 107–109: the authors should comment on the impact of DAPT in the neurons. While their interpretation will not change, as they only present control data, it is a useful piece of information for the reader.

Thank you for this comment. In line with Reviewer #1's feedback, we have expanded our explanation of the effects of DAPT.

Line 113: In Figures 1c–f, it would be good to specify if the larvae are wild-type or DAPT-treated.

*Thank you for this. We added the explanation, "Notably, we found that the *tph*-positive cells were clearly separated into two distinct clusters on the UMAP (Fig. 1c,d), which shows only control cells extracted from the integrated dataset of control and DAPT-treated samples".*

Line 140: How were the "regions" compared? Do the authors mean cells? Or clusters? Please be more precise.

We fixed it to "gene expression profiles", in the process of addressing the final concern of this reviewer (see below).

*Lines 164–166: The authors claim that *rx* is expressed in *Opn5L*-expressing cells and reference Figure 2h, but in this Figure *Opn5L* expression is not shown.*

*Thank you for this comment. We agree that in the original version, *rx* (as well as other genes) was shown only by single *in situ* hybridization, which did not directly demonstrate co-localized expression with *opn5L/z167*-positive cells. To address this, we have added double *in situ* hybridization with *z167* in the revised version (Figure 3 in the revised manuscript). In addition, we have updated the text to refer to *opn5L/z167*-expressing cells instead of *opn5L*-expressing cells alone. The co-localization of *opn5L* and *z167* is shown in Figure 2.*

In Figure 2, in general, the authors should present double stains with tph and/or opn5L.

Thank you for this comment. We made new Figure 3 to show the double *in situ* hybridization for genes expressing at the critical site. As well, we showed *tph* and *z167* double *in situ* hybridization in Figure 2, as suggested.

Lines 204: There is no reference for this “longstanding view”

Thank you for pointing this out. We added two references in the sentence.

Line 300: What do the authors mean by “this study”? This study doesn’t show this.

Thank you for this. The reviewer is correct. We deleted it.

Reference 3 is incorrect. The title of the paper has changed in Science Advances.

Thank you for this. We fixed it.

Video S1 is not really supportive of migration. Same for Figure 3e–e’.

See above (c)(d).

For Table 1, very minimal information is provided as to how these genes were identified and how the genes to be tested were selected.

Thank you for this comment. The reviewer is correct. We fixed the sentences and make it clear how we prioritized these candidate genes. “Based on the scRNA-seq data, we compared the gene expression profiles of all cells within the two *tph*-expressing clusters: one corresponding to dorsal serotonergic neurons (orange-lined rectangle in Fig. 1d,e) and the other to anterior serotonergic neurons (light blue-dot-lined rectangle in Fig. 1d,e). Among the genes enriched in the dorsal *opn5L*-positive cluster, we excluded hypothetical proteins, general cytoskeletal components, and those with unknown function, and prioritized candidates with a high likelihood of involvement in neurogenesis or neuronal function. These included transcription factors, signaling molecules, and membrane proteins, many of which have been previously implicated in neural development^{e.g.,3,10,18,43} (highlighted in orange) and validated their expression patterns through *in situ* hybridization. “

”

REVIEWERS' COMMENTS

Reviewer #1 (Remarks to the Author):

The authors have addressed all of my concerns and I now recommend this study for publication.

Reviewer #2 (Remarks to the Author):

All the reasonable questions posed by the reviewers have been answered comprehensively. I have no additional suggestions to offer.

Reviewer #3 (Remarks to the Author):

The authors have addressed the reviewers' comments successfully and, in my opinion, this led to the improvement of the manuscript. A couple concerns remain:

- Based on Supplementary Figure S2, *Opn5L* is also expressed in other cells besides the dorsal serotonergic neurons. How do the authors know that the effect of the *Opn5L* knock-down is because of its absence in the dorsal serotonergic neurons and not in the other cells? On that note, it would be good to highlight what kind of cells these are.

Thank you for this comment. Since Figure S2 shows an integrated UMAP of 24–96 h, it is difficult to clearly distinguish time-dependent expression patterns and spatial domains. The actual expression domains of *Opn5L* at each developmental stage can be seen in the *in situ* results in Figure S3. As shown there (and clearly noted in the figure legend), *Opn5L* is broadly expressed in the ectoderm from 12–30 h, with enrichment in the region predicted to give rise to dorsal serotonergic neurons. However, after 36 h its expression becomes restricted to the dorsal serotonergic neuron domain, as confirmed by both the detailed single-cell data and the *in situ* results in Figure S3.

Regarding the question of whether the knockdown of *Opn5L* could also affect the early expression before 30 h, we cannot completely rule out this possibility. However, the knockdown experiments were analyzed at 4 days, and it is unlikely that *Opn5L* mRNA expressed before 30 h would retain functional activity until

that time point. Moreover, since the embryos developed normally up to day 4, we consider that the influence of early *Opn5L* knockdown was not substantial in the present experiments.

- I am still not convinced that Figure 4 shows migration. While the movie is much better now and indeed shows movement, migration is an active, energy-dependent process. I am even more concerned as they use this movement to report “the first neuronal migration in *Ambulacraria* reshaping our understanding of deuterostome nervous system evolution”. I still believe that with the current data, no neuronal migration has been shown.

We understand the reviewer’s concerns. Since our model organism does not allow for complete knock-in experiments, we acknowledge that all of the data presented might be indirect. Nevertheless, we believe that several observations support the migration of dorsal serotonergic neurons: live imaging of fluorescent protein–labeled cells driven by gene regulatory elements specific to dorsal serotonergic neurons, the close approach of two serotonergic neurons, and the behavior of neuronal cells delaminating into the blastocoel. At the same time, as stated in the main text, it is indeed true that we could not directly demonstrate this process because knockdown validation was not possible due to the early expression of *sna*, and complete knock-in–based fluorescent protein labeling and live imaging of neurons could not be performed. Therefore, we have decided to move the images (originally Figure 4e,e’) from the Supplementary movie to Supplementary Figure 6 and delete the sentence beginning with “*This marks the first report ...*”, as highlighted by the reviewer. In addition, we have softened the preceding sentence by revising “*Our data provide evidence that ...*” to “*Our data provide the possibility that ...*”.

Minor comments:

- I think that the phrase “and enhanced detection of rare neuronal populations by inhibiting Delta-Notch signaling with DAPT” in the Abstract is an unnecessary detail and can be confusing for the reader.

We agree with this comment. We deleted it.

- Line 163: “expressing” should probably be “expression in”

Thank you for this. We fixed it to “expressed in”.

- In Figures 2k,l, why don't the authors co-stain with ebf3 to show beyond doubt that these are the anterior serotonergic neurons?

Thank you for this comment. We added the double staining data for ebf3 and tph, serotonin synthase, which indicates ebf3-expressing cells are serotonergic neurons.

- Line 223: "functioned" should be "function" or "functional"

We fixed it.

- Line 275: "overlapped" should be "overlapping"

Thank you for this point. We fixed it together with other overlapped in the text.